# Heat Shock Factor 1 forms nuclear condensates and restructures the yeast genome before activating target genes

Linda S Rubio, Suman Mohajan, David S Gross*

Department of Biochemistry and Molecular Biology Louisiana State University Health Sciences Center, Shreveport, United States

**\*For correspondence:**
david.gross@lsuhs.edu

**Competing interest:** The authors declare that no competing interests exist.

**Abstract** In insects and mammals, 3D genome topology has been linked to transcriptional states yet whether this link holds for other eukaryotes is unclear. Using both ligation proximity and fluorescence microscopy assays, we show that in *Saccharomyces cerevisiae*, *Heat Shock Response* (*HSR*) genes dispersed across multiple chromosomes and under the control of Heat Shock Factor (Hsf1) rapidly reposition in cells exposed to acute ethanol stress and engage in concerted, Hsf1-dependent intergenic interactions. Accompanying 3D genome reconfiguration is equally rapid formation of Hsf1-containing condensates. However, in contrast to the transience of Hsf1-driven intergenic interactions that peak within 10–20 min and dissipate within 1 hr in the presence of 8.5% (v/v) ethanol, transcriptional condensates are stably maintained for hours. Moreover, under the same conditions, Pol II occupancy of *HSR* genes, chromatin remodeling, and RNA expression are detectable only later in the response and peak much later (>1 hr). This contrasts with the coordinate response of *HSR* genes to thermal stress (39°C) where Pol II occupancy, transcription, histone eviction, intergenic interactions, and formation of Hsf1 condensates are all rapid yet transient (peak within 2.5–10 min and dissipate within 1 hr). Therefore, Hsf1 forms condensates, restructures the genome and transcriptionally activates *HSR* genes in response to both forms of proteotoxic stress but does so with strikingly different kinetics. In cells subjected to ethanol stress, Hsf1 forms condensates and repositions target genes before transcriptionally activating them.

## eLife assessment

This is a **valuable** contribution to our understanding of how different cell stressors (ethanol or heat-shock) elicit unique responses at the genomic and topographical level under the regulation of yeast transcription factor Hsf1, providing **solid** evidence documenting the temporal coupling (or lack thereof) between Hsf1 aggregation and long-range communication among co-regulated heat-shock loci versus chromatin remodeling and transcriptional activation. A particular strength is the combination of genomic and imaging-based experimental approaches applied to genetically engineered in vivo systems.

## Introduction

Genomes of higher eukaryotes are organized into multiple hierarchical levels. Chromosomes are segregated within individual territories, and within chromosomal territories active and inactive regions are separated into topologically associated domains (TADs; *Wendt and Grosveld, 2014*). Within TADs, DNA loops are formed to permit the interaction between enhancers or silencers and the promoters of their target loci. Although this hierarchy suggests a static view of the genome, recent studies have suggested that the genome is quite dynamic, as multiple points of interaction form in response

to developmental cues and other stimuli. This dynamic restructuring ranges from the interactions between co-regulated genes (*Fanucchi et al., 2013*; *Papantonis et al., 2012*; *Park et al., 2014*; *Schoenfelder et al., 2010*), to the convergence of enhancers dispersed across multiple chromosomes into a hub that regulates a single gene (*Monahan and Lomvardas, 2015*), to the reorganization of the genome that occurs in the zygote (*Schulz and Harrison, 2019*).

Despite its evolutionary distance, the yeast *Saccharomyces cerevisiae* also possesses an organized genome. Centromeres are located in a cluster at the spindle pole body, while chromosomal arms are extended with telomeres and the nucleolus located at the opposite side of the nucleus (*Duan et al., 2010*), resulting in a Rabl-like configuration (*Taddei and Gasser, 2012*). And as in higher eukaryotes, the budding yeast genome is organized into TAD-like structures (*Eser et al., 2017*) subdivided into smaller loop domains (*Hsieh et al., 2015*). Also as is the case in mammalian cells, the yeast genome is not static. Genes, including *INO1*, *GAL1*, and *GAL10*, have been observed to reposition from the nuclear interior to the nuclear periphery upon their activation (*Brickner et al., 2019*; *Brickner and Walter, 2004*; *Cabal et al., 2006*; *Casolari et al., 2004*; *Dieppois et al., 2006*; *Green et al., 2012*). In response to methionine starvation, several Met4-regulated genes have been observed to cluster (*Du et al., 2017*; *Lee et al., 2024*).

More dramatic still are *Heat Shock Response* (*HSR*) genes under the regulation of Heat Shock Factor 1 (Hsf1). Upon exposure to acute thermal stress (heat shock [HS]), *HSR* genes dispersed across multiple chromosomes transcriptionally activate and engage in novel *cis*- and *trans*-intergenic interactions between one another – principally although not exclusively involving coding regions – that culminate in their coalescence into intranuclear foci (*Chowdhary et al., 2017*; *Chowdhary et al., 2019*). These stress-induced foci, comprised of Hsf1 and components of the transcriptional machinery, form quickly, rearrange rapidly and dissolve suddenly (*Chowdhary et al., 2022*). As such, Hsf1 clusters exhibit properties of biomolecular condensates, defined as self-organized membrane-free compartments enriched in specific macromolecules that may or may not be phase-separated (*Banani et al., 2017*). Hsf1 condensate formation elicited by heat shock, like that of *HSR* gene interactions, parallels the kinetics of induction and attenuation of Hsf1-dependent genes (*Chowdhary et al., 2017*; *Chowdhary et al., 2022*).

The Hsf1-driven heat shock response is a fundamental, evolutionarily conserved transcriptional program characterized by the gene-specific transcription factor (TF) Hsf1, its DNA recognition element (heat shock element [HSE]) and a core set of target genes encoding molecular chaperones and co-chaperones (reviewed in *Verghese et al., 2012*). In absence of proteotoxic stress, yeast Hsf1 is bound by Hsp70 and its co-chaperone Sis1 in the nucleoplasm (*Feder et al., 2021*; *Krakowiak et al., 2018*; *Peffer et al., 2019*; *Zheng et al., 2016*). Upon encountering stress, Hsp70 is titrated by unfolded proteins, particularly orphan ribosomal proteins located in the nucleolus and nascent polypeptides in the cytosol (*Albert et al., 2019*; *Ali et al., 2023*; *Tye et al., 2019*; *Tye and Churchman, 2021*). This results in the release of Hsf1 which then trimerizes and binds to HSEs located upstream of ~50 genes whose transactivation is dependent on this factor (*Pincus et al., 2018*). Once proteostasis is reestablished, excess Hsp70 binds Hsf1, inactivating it, thereby closing the negative feedback loop that regulates Hsf1 transcriptional activity (*Krakowiak et al., 2018*; *Zheng et al., 2016*). Note that in yeast, a fraction of Hsf1 is constitutively trimeric and it is this species that binds high-affinity HSEs even under non-stressful conditions (*Giardina and Lis, 1995*; *Pincus et al., 2018*).

In addition to thermal stress, Hsf1 can be activated by chemical stressors such as ethanol. Ethanol is a metabolite of glucose breakdown that budding yeast cells secrete into their surroundings. Ethanol production helps yeast outcompete other microbes in the environment (*Liti, 2015*; *Piskur et al., 2006*; *Rozpędowska et al., 2011*). Furthermore, once glucose is depleted, ethanol serves as an alternative carbon source (*Piskur et al., 2006*). Given this strategic use of ethanol, it is of fundamental importance for yeast to have a mechanism in place to respond to the stress that ethanol elicits. Similar to thermal stress, exposure to ethanol causes a large number of cellular perturbations including disruption of the plasma membrane (*Piper et al., 1994*); disruption of the H⁺ ATPase and intracellular acidification (*Rosa and Sá-Correia, 1991*; *Rosa and Sá-Correia, 1996*; *Triandafillou et al., 2020*); production of reactive oxygen species (*Bandas and Zakharov, 1980*; *Davidson et al., 1996*); depolymerization of the actin cytoskeleton (*Homoto and Izawa, 2018*; *Tan et al., 2017*); cell cycle arrest (*Johnston and Singer, 1980*; *Kubota et al., 2004*); disruption of mRNP transport to the daughter cell and formation of stress granules (*Grousl et al., 2009*; *Kato et al., 2011*); global inhibition of transcription and

translation (*Bresson et al., 2020*; *Gasch et al., 2000*); and formation of protein aggregates (*Piper, 1995*; *Plesset et al., 1982*; *Stanley et al., 2010*). The cell counteracts many of these perturbations through the enhanced production of molecular chaperones (*Desroches Altamirano et al., 2024*; *Verghese et al., 2012*).

Here, we investigate activation of the HSR in yeast exposed to 8.5% ethanol (ethanol stress [ES]) and compare it to the response induced by exposure to 39°C (HS). We find that like HS, exposure to ES induces transcription of Hsf1-regulated *HSR* genes and induces concerted intergenic interactions between them. In contrast to HS, however, Hsf1 condensate formation and *HSR* intergenic interactions cells peak well before transcription. The delay in transcriptional induction may be linked to a widespread increase in nucleosome occupancy that occurs upon exposure of cells to ethanol. At longer times of ethanol exposure, *HSR* gene transcript accumulation continues to increase while *HSR* gene coalescence has already dissipated. Likewise, ES-induced Hsf1 condensates are present for a prolonged period, in contrast to HS-induced condensates that begin to dissipate shortly after they appear. Collectively, our data indicate that different stimuli drive distinct transcription, chromatin, topologic and condensation phenomena, yet all are dependent on Hsf1.

## Results

### Thermal stress and ethanol stress elicit distinct proteotoxic responses in yeast

*Saccharomyces cerevisiae* in the wild metabolizes glucose and other sugars into ethanol, which the yeast secretes into the environment to suppress microbial competition. Therefore, it is likely that yeast has evolved mechanisms to contend with ethanol toxicity. Indeed, a common laboratory strain (W303) retains viability when cultivated in the presence of 8.5% ethanol, although its ability to proliferate is diminished (*Figure 1A and B*). As assessed by the presence of Hsp104-containing foci (a measure of protein aggregation; *Liu et al., 2010*), the rate of cytosolic protein aggregation is similar in cells exposed to either 8.5% ethanol or 39°C thermal stress (*Figure 1D and E*; see *Figure 1C* for experimental design). However, it is notable that the number and volume of Hsp104 foci, and by extension protein aggregation, is substantially higher in cells subjected to ethanol stress.

We next investigated the subcellular localization of the Hsp70 co-chaperone, Sis1, in cells exposed to ethanol versus thermal stress. Previous work has shown that Sis1 is diffusely localized within the nucleus under control conditions where it promotes binding of Hsp70 to Hsf1, thereby repressing the HSR (*Feder et al., 2021*). In response to acute HS, Sis1 relocates to nucleolus where it forms a perinucleolar ring, and to the cytoplasm where it forms cytosolic clusters that colocalize with Hsp104 and spatially associate with the endoplasmic reticulum (*Feder et al., 2021*). Consistent with these previous observations, we observed rapid relocalization of Sis1 in response to a 39°C HS. It formed a ring-like structure within 2.5 min consistent with a perinuclear location. Such relocalization lasted at least 60 min and spatially separated Sis1 from Hsf1 (*Figure 2A and B*). A similar result was obtained when cells were exposed to 42°C. In response to ES, Sis1 remained largely co-localized with Hsf1 within the nucleoplasm although some enrichment at the nuclear periphery was evident. This was the case whether cells were exposed to 5% or 8.5% ethanol (*Figure 2A and B*). In addition, a 60 min exposure to 8.5% ethanol resulted in the formation of prominent cytosolic Sis1 puncta that were less evident in the HS sample (*Figure 2A*). These observations suggest that these two stressful treatments elicit a qualitatively distinct response. Experiments described below provide further support for this possibility.

### Ethanol stress induces transcriptional activation of Hsf1-dependent genes but with delayed kinetics and reduced expression compared to thermal stress

To gain further insight into the mechanism by which *S. cerevisiae* contends with ethanol-induced proteotoxicity, we assessed the kinetics of transcriptional activation of *HSR* genes in cells exposed to ethanol stress. Cells were cultivated to early log phase in rich YPD medium, then ethanol was added to a final concentration of 8.5% and cell aliquots were removed at 0-, 10-, 20-, and 60 min. Transcription was terminated through addition of sodium azide (see Materials and methods). A parallel culture was

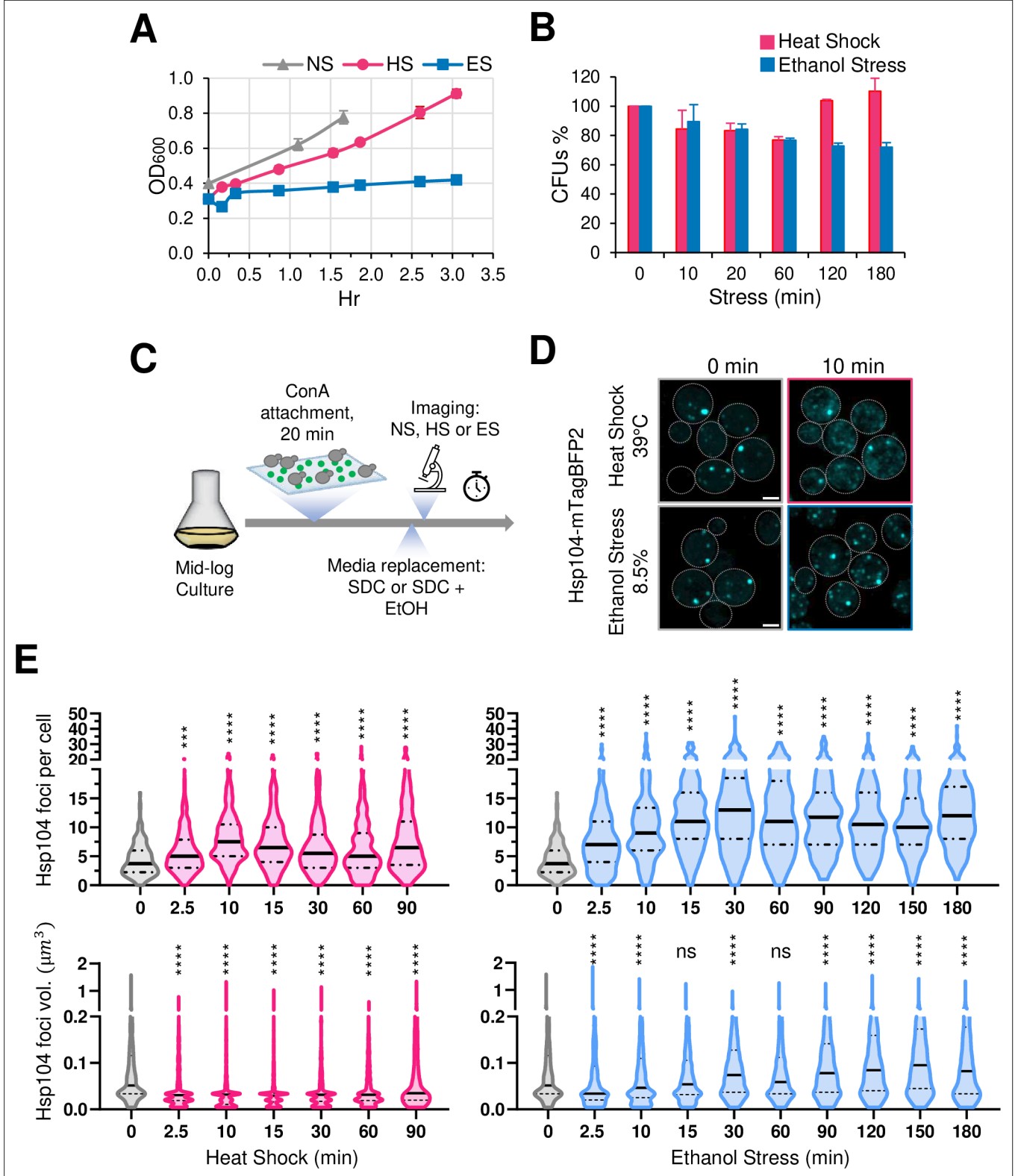

**Figure 1.** Thermal and chemical stresses used in this study elicit distinct proteotoxic responses. (**A**) Growth curve of strain W303-1B grown in liquid culture (YPDA). Mid-log phase cultures were diluted to $OD_{600}$=0.4 and shifted to different conditions: no stress (NS, 25°C), heat shock (HS, 39°C), or ethanol stress (ES, 8.5% v/v, 25°C). $OD_{600}$ was monitored over time. Means and SD are shown. N=2. (**B**) Viability assay of W303-1B cells following exposure to heat shock (25° to 39°C upshift for the indicated time) or ethanol stress (8.5% v/v ethanol for the indicated time at 25°C). An aliquot was taken from each condition at the indicated stress timepoints and diluted in rich media. Cells were spread on YPDA plates and grown at 30°C for

*Figure 1 continued on next page*

*Figure 1 continued*

3 days. Colony forming units (CFUs) were determined using ImageJ/FIJI. Plotted are percentages of CFUs of stressed cells normalized to those of the 0 min control. Graphs depict means + SD. N=2. (**C**) Experimental strategy for imaging Hsp104 foci. Cells were attached to a concanavalin A (ConA)-coated surface, followed by heat shock or ethanol stress treatment (see Materials and Methods). Synthetic complete media (SDC) was supplemented with ethanol to a final concentration of 8.5% for ES samples. Scale bar: 2 µm. (**D**) Both heat shock and ethanol stress induce formation of Hsp104 foci. DPY1561 haploid cells were attached to a VAHEAT substrate using Concanavalin A and subjected to an instantaneous heat shock (25° to 39°C) or to ethanol stress (25°C, 8.5% v/v). The 0 min control was kept at 25°C without stress. Hsp104-mTagBFP2 foci were visualized by confocal microscopy. Shown are maximal projections of 11 z-planes, taken with an interplanar distance of 0.5 µm. Scale bar: 2 µm. (**E**) Cells subjected to the above treatments were assayed for Hsp104 puncta number and volume. Violin plots summarizing this analysis are depicted. An average of 200 cells per timepoint per condition was quantified using Imaris image analysis software (v.10.0.1). For this analysis, we made the assumption that the diffuse Hsp104 clusters seen in HS cells are comparable to the compact Hsp104 foci in ES cells. N=2. Significance was determined by Mann Whitney test, stress vs. no stress (0 min). ***, $p<0.001$; ****, $p<0.0001$; ns, not significant.

The online version of this article includes the following source data for figure 1:

**Source data 1.** Spreadsheet tabulates the number of Hsp104 foci per cell and the volume of individual Hsp104 foci (*Figure 1E*).

exposed to an instantaneous 30° to 39°C heat shock and cells were removed at the corresponding time points. Transcription was terminated as above.

While cells exposed to heat shock displayed a rapid and substantial increase in *HSR* mRNA levels (typically >10 fold increase within 10 min of thermal upshift), those exposed to ethanol stress only weakly induced the same cohort of genes (*Figure 3A*). However, while HS induced a transient increase in RNA expression, ES induced a sustained increase that was evident at all Hsf1-dependent genes tested (see also *Figure 3—figure supplements 1 and 2*). A corresponding immunoblot analysis of two Hsf1 targets, Hsp104 and Btn2, strengthens the notion that HS induces a rapid yet transient response, whereas ES induces a delayed, yet far more sustained increase in gene expression (*Figure 3—figure supplement 3*). In the case of *HSP12* and *HSP26*, whose transcription is under the dual regulation of Msn2 and Hsf1, exposure to heat shock resulted in a high level of induction as previously observed (*Chowdhary et al., 2019*) yet exposure to 8.5% ethanol failed to cause detectable activation during the first 20 min (*Figure 3B*; *Figure 3—figure supplement 1*). Nonetheless, as described below, both *HSP12* and *HSP26* respond to acute ethanol stress but do so through their inducible and dramatic 3D genomic repositioning.

## Pol II recruitment and histone eviction are delayed in ethanol-stressed cells and this correlates with a transient, widespread increase in nucleosome density

The delayed transcriptional response of *HSR* genes in ES- versus HS-treated cells prompted us to investigate occupancy of Hsf1, RNA Pol II and histones at these genes over a time course. A possible explanation for the delay in activation in cells exposed to ethanol stress is less rapid (or reduced) Hsf1 binding to the genes' upstream regulatory regions. To explore this possibility, we exposed cells to either thermal or chemical stress and processed them for chromatin immunoprecipitation (ChIP) analysis. As previously observed (*Kim and Gross, 2013*; *Pincus et al., 2018*; *Sekinger and Gross, 2001*), occupancy of Hsf1 at its target loci increases many-fold following a brief heat shock (*Figure 4B*, left, dark red; see *Figure 4A* for location of primers). Hsf1 occupancy typically declines after 60 min of continuous thermal stress and in the case of *TMA10*, dissociation begins much sooner. In response to ethanol stress, Hsf1 occupancy steadily increased, in most cases reaching maximal levels by 20 min and plateauing thereafter (*Figure 4B*, left, black). These results suggest that ethanol stress induces binding of Hsf1 to its cognate HSEs to a degree similar to heat shock, yet such binding is more gradual. Moreover, given the above, Hsf1's binding to these sites fails to elicit a corresponding transcriptional response.

In light of this disconnect between Hsf1 binding and *HSR* mRNA production, we evaluated abundance of the large subunit of Pol II at representative genes over the same time course. In response to heat shock, Rpb1 is rapidly recruited to the promoters and coding regions of each *HSR* gene, peaking within 2.5 min and then gradually declining over the next ~60 min (*Figure 4B*, middle, red and pink traces). By contrast, its occupancy is noticeably delayed in cells exposed to ethanol stress (blue traces), consistent with reduced transcript levels (*Figure 4—figure supplement 1B*). The increase in *HSR* mRNA in heat-shocked cells parallels, yet consistently lags, the abundance of Pol II within *HSR* gene

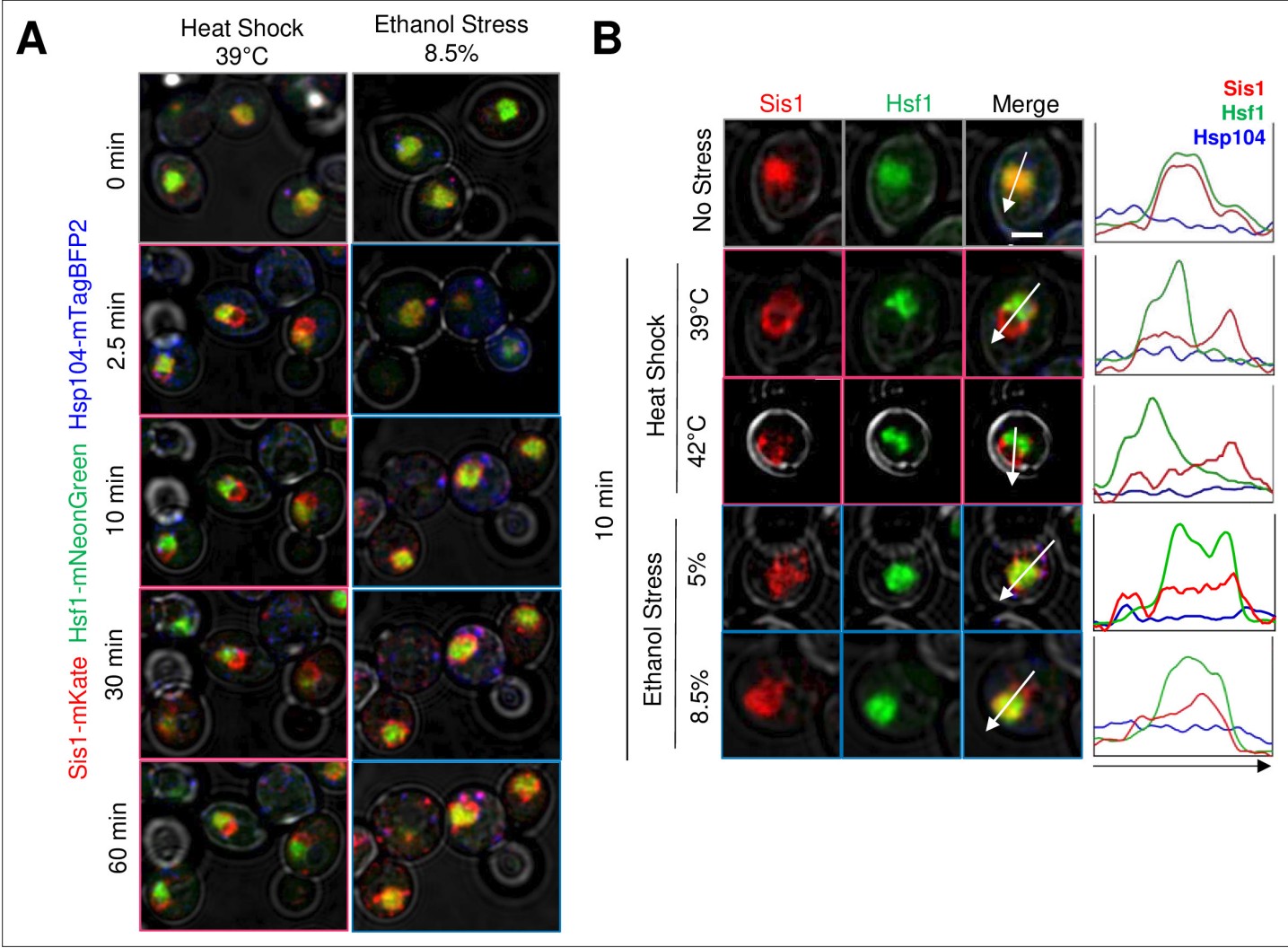

**Figure 2.** Heat shock and ethanol stress elicit distinct patterns of Sis1 subcellular relocalization. (**A**) Live cell confocal microscopy of the diploid strain LRY033 expressing Sis1-mKate, Hsf1-mNeonGreen, and Hsp104-mTagBFP2. Cells were treated as in *Figure 1D*. 11 z-planes were captured with an interplanar distance of 0.5 µm. Shown is a representative plane for each timepoint. (**B**) Subcellular localization analysis of Sis1, Hsf1, and Hsp104 in cells subjected to no stress (25°C), heat shock (at 39° or 42°C), or ethanol stress (at 5% or 8.5% v/v [25°C]) for 10 min. Cells from strain LRY033 were treated as described in *Figure 1D*. A representative plane is shown for each condition. Line profiles are plotted for each channel on the right. Arrows were drawn to bisect the nucleus. Scale bar: 2 µm.

coding regions (*Figure 4—figure supplement 1A*). This delay in reaching peak accumulation may reflect contributions beyond RNA synthesis, such as transient enhanced stability of *HSR* transcripts during the acute phase of heat shock.

To obtain further insight into the chromatin landscape present during the two stresses, we assayed histone H3 abundance as a measure of nucleosome density. In response to heat shock, nucleosomes are rapidly displaced over the promoter and coding regions of the strongly expressed *HSR* genes. Following this initial phase (~20 min), nucleosomes reassemble over these genes and in certain cases return to their original density by 60 min (*Figure 4B*, right, red and pink traces). This dramatic and dynamic remodeling has been previously observed (*Kremer and Gross, 2009*; *Zhang et al., 2014*; *Zhao et al., 2005*). In contrast, ethanol stress elicits a transient increase in H3 abundance over all five genes (*Figure 4B*, right, blue traces). This apparent increase in chromatin compaction is not restricted to *HSR* genes; a variety of unrelated loci, including a stress-responsive, Msn2-regulated gene (*PGM2*), two constitutively expressed genes (*TUB1, ACT1*), two genes assembled into *SIR*-dependent heterochromatin (*HMLα1, YFR057w*) and a non-transcribed region (*ARS504*) also exhibit a transient increase in nucleosome density in ethanol-exposed cells (*Figure 4—figure supplement 2A*).

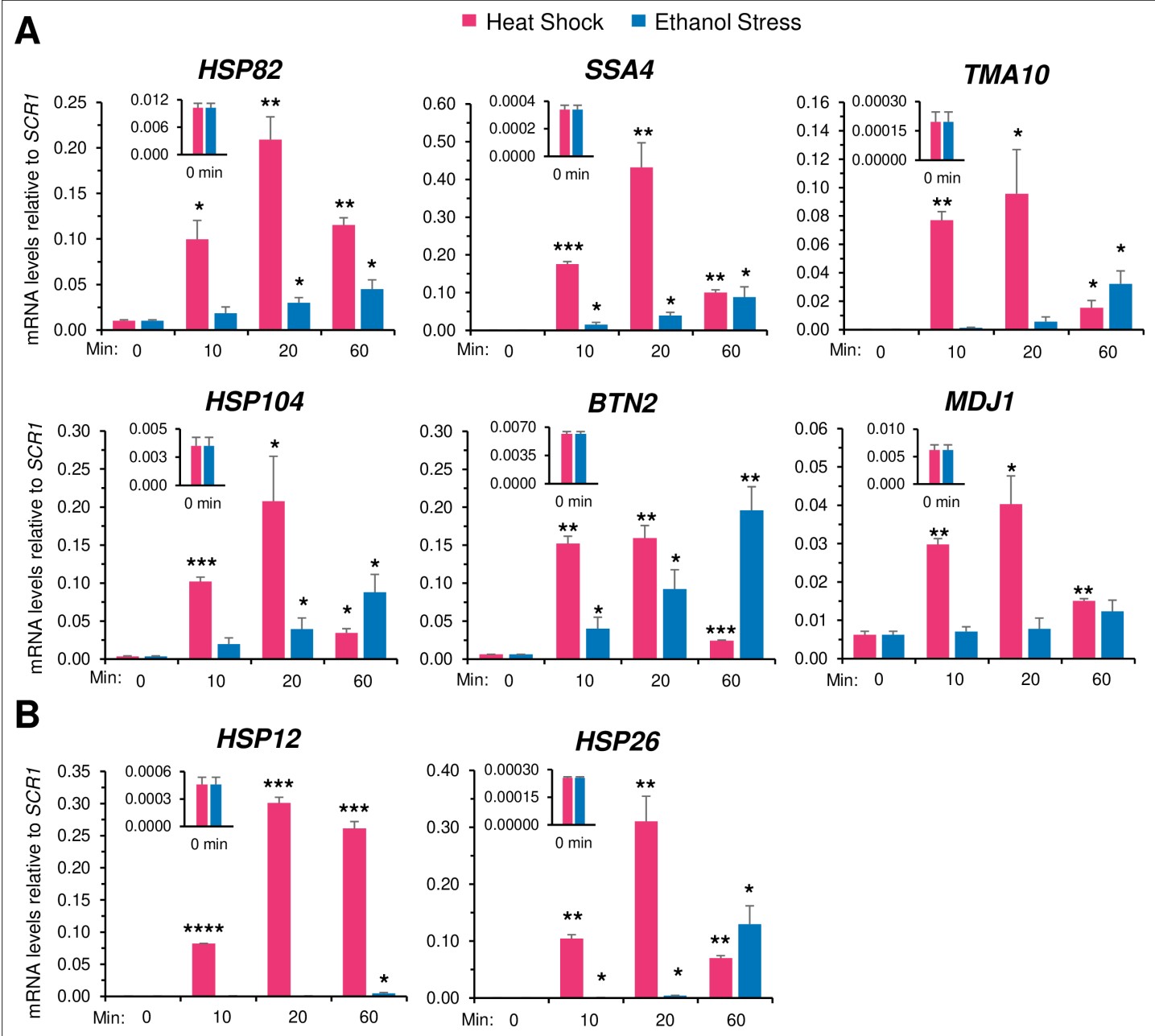

**Figure 3.** Ethanol stress transcriptionally induces *Heat Shock Response (HSR)* genes but with markedly slower kinetics than thermal stress. (**A**) RNA abundance of Hsf1-dependent *HSR* genes was determined by Reverse Transcription-qPCR in strain W303-1B. Heat shock was performed at 39°C; ethanol stress was done using 8.5% (v/v) ethanol at 25°C. Insets display transcript abundance using a zoomed-in scale. Depicted are means + SD. N=2, qPCR = 4. Statistical analysis: T-test, one-tailed, no stress vs. stress conditions. *, $p<0.05$; **, $p<0.01$, ***; $p<0.001$. (**B**) As in (**A**), but the Hsf1-, Msn2-dual regulated genes *HSP12* and *HSP26* were evaluated. *, $p<0.05$; ***, $p<0.001$; ****, $p<0.0001$.

The online version of this article includes the following source data and figure supplement(s) for figure 3:

**Source data 1.** Spreadsheet contains the RT-qPCR data plotted for *HSR* RNA analysis (*Figure 3*).

**Figure supplement 1.** Ethanol stress transcriptionally induces *HSR* genes.

**Figure supplement 1—source data 1.** Spreadsheet contains mRNA levels for ethanol stressed cells (*Figure 3—figure supplement 1A, B*), as well as *SCR1* RNA levels for heat shock and ethanol stress (*Figure 3—figure supplement 1C*).

**Figure supplement 2.** In response to chronic heat stress *HSR* mRNA levels gradually attenuate, whereas in response to chronic ethanol stress they remain constant.

**Figure supplement 2—source data 1.** Spreadsheet contains mRNA levels for *HSR* genes under chronic heat shock and ethanol stress (4 hr; *Figure 3—*

*Figure 3 continued*

figure supplement 2).

**Figure supplement 3.** Induced HSR protein production is evident early during heat shock while it is delayed during ethanol stress.

**Figure supplement 3—source data 1.** Raw files for Hsp104, Btn2, and histone H3 blots (*Figure 3—figure supplement 3A*).

**Figure supplement 3—source data 2.** Labeled blots for Hsp104, Btn2, and histone H3 protein quantification (*Figure 3—figure supplement 3A*).

**Figure supplement 3—source data 3.** Spreadsheet tabulates Hsp104 and Btn2 protein levels normalized to histone H3 (*Figure 3—figure supplement 3B*).

In agreement with this idea, chromatin volume (as assessed by the signal arising from an H2A-mCherry protein fusion) decreases upon exposure of cells to ethanol stress (and much more transiently to heat stress; *Figure 4—figure supplement 2*[B, C]). These data suggest that the increase in nucleosome density antagonizes Pol II recruitment and its subsequent release into the coding regions of *HSR* genes in ES cells. Once Pol II has been stably recruited, elongation and concomitant histone eviction ensues (*Figure 4B*, light blue). Altogether, our ChIP data indicate that Hsf1 binds to its target enhancers less readily in ethanol-stressed than in thermally stressed cells. This impediment to Hsf1 occupancy is magnified by a corresponding, and more severe, hindrance to Pol II recruitment resulting in a pronounced delay in *HSR* gene transcription.

## Acute ethanol stress induces rapid and profound 3D genomic repositioning of *HSR* loci

An intriguing feature of *HSR* genes is the fact that they coalesce into discrete intranuclear foci in response to heat shock. Such interactions have been documented using both molecular (Chromosome Conformation Capture [3C]) and imaging (fluorescence microscopy) approaches (*Chowdhary et al., 2017*; *Chowdhary et al., 2019*; *Chowdhary et al., 2022*; *Rubio and Gross, 2023*). Moreover, such physical interactions specifically involve Hsf1 targets irrespective of their location in the genome. Other loci, including adjacent, transcriptionally active genes, show little or no tendency to interact with Hsf1-dependent genes. Such *cis*- and *trans*-interactions principally involve gene coding regions and are highly dynamic, typically peaking at 2.5 min and dissipating by 30- to 60 min. The kinetics of HS-induced coalescence often, although not always, correlate with kinetics of transcriptional induction; they also parallel the formation of Hsf1 condensates (*Chowdhary et al., 2017*; *Chowdhary et al., 2019*; *Chowdhary et al., 2022*) as discussed further below.

Given these previous observations, we wished to know if ethanol stress induced similar 3D genome restructuring. It seemed unlikely that such topological changes would occur during the initial phase of ES since only weak Pol II occupancy and low *HSR* transcript levels are observed (*Figures 3 and 4*). However, as shown in *Figure 5*, exposure to 8.5% ethanol triggered frequent intergenic interactions between Hsf1 targets during the first 10 min as revealed by Taq I-3C, a highly sensitive, quantitative version of 3C (*Chowdhary et al., 2020*) (see *Figure 5—figure supplement 1* for location of 3C primers). Both intra- and interchromosomal interactions can be detected. Moreover, the interaction frequencies following this exposure in most cases equaled, and in some instances exceeded, those detected in cells heat-shocked for 2.5 min (*Figure 5A and B*), when peak 3C interactions occur in thermally stressed cells (*Chowdhary et al., 2017*). A detailed kinetic analysis revealed that intergenic interactions elicited by ethanol stress, similar to those induced by thermal stress, are highly dynamic: detectable within 2.5 min, peak shortly thereafter (within 10–20 min) and largely attenuate by 60 min (*Figures 5 and 6A*).

It has been previously suggested that a functional link exists between gene looping and transcriptional activation (*Ansari and Hampsey, 2005*; *O'Sullivan et al., 2004*). Indeed, gene loops and other intragenic interactions (*Chowdhary et al., 2017*) are readily detected within *HSR* genes in cells exposed to ethanol. However, as is the case with intergenic interactions, these topological changes are kinetically uncoupled from both transcription and Pol II occupancy. They are detected quite early (within 2.5 min), peak soon thereafter (at 10 min) and attenuate by 60 min (*Figure 6B* and *Figure 5—figure supplement 2*). Taken together, our 3C and expression analyses indicate that 3D genomic repositioning and intragenic looping of *HSR* genes precedes the maxima of transcript levels and Pol II occupancy. Moreover, for certain loci (e.g. *HSP12* and *HSP26*), they argue that neither transcription

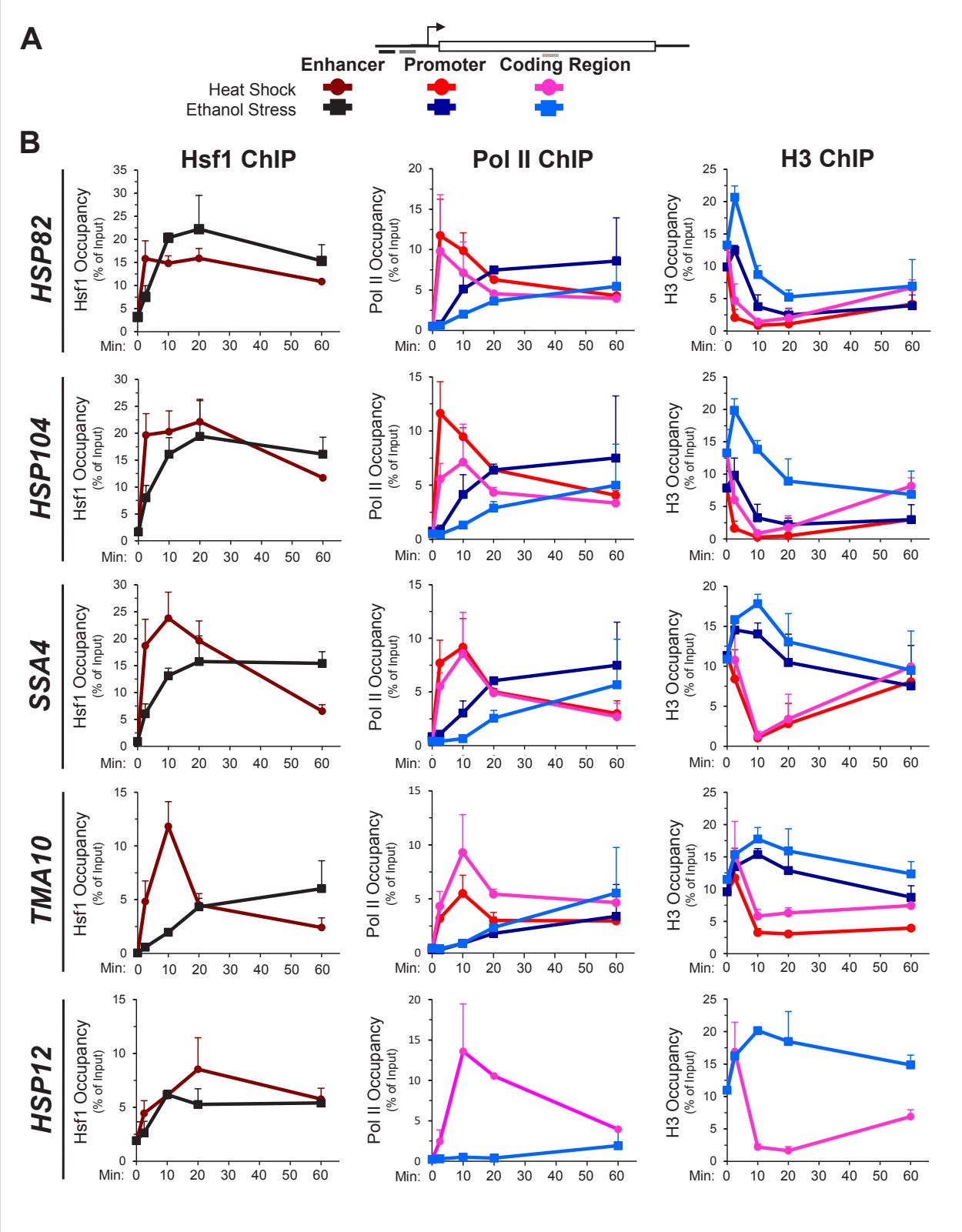

**Figure 4.** Hsf1 and Pol II recruitment to *HSR* genes is delayed in ethanol stressed cells, while histone occupancy transiently increases. (**A**) Map of a representative *HSR* gene depicting locations of primers used for chromatin immunoprecipitation (ChIP) analysis. Heat shock: shades of red and pink. Ethanol stress: shades of blue. (**B**) ChIP analysis of Hsf1, Pol II (Rpb1) and histone H3 occupancy to the enhancer (UAS), promoter and coding regions of the indicated genes. Mid-log cultures of strain BY4741 were subjected to the indicated times of heat shock (39°C) or ethanol stress (8.5% v/v, 25°C).

*Figure 4 continued on next page*

*Figure 4 continued*

Time points evaluated for all three factors: 0-, 2.5-, 10-, 20-, and 60 min. Antibodies raised against full-length Hsf1, CTD of Rbp1 or the globular domain of histone H3 were used (see Materials and methods). ChIP signals were normalized to input. Shown are means + SD. N=2, qPCR = 4.

The online version of this article includes the following source data and figure supplement(s) for figure 4:

**Source data 1.** Spreadsheet tabulates Hsf1, Pol II, and histone H3 occupancy.

**Figure supplement 1.** Pol II occupancy of the indicated *HSR* genes vs. their mRNA levels in cells subjected to either heat shock or ethanol stress.

**Figure supplement 1—source data 1.** Spreadsheet tabulates Pol II occupancy vs. *HSR* mRNA levels in cells exposed to heat shock or ethanol stress (*Figure 4—figure supplement 1*).

**Figure supplement 2.** Ethanol and thermal stresses induce chromatin compaction.

**Figure supplement 2—source data 1.** Histone H3 ChIP analysis of different genomic loci, under HS or ES (*Figure 4—figure supplement 2A*).

---

nor Pol II recruitment is required to drive 3D topological changes in these genes. Further evidence in support of this is provided below.

## Live cell imaging reveals that *HSR* genes coalesce to a similar degree under ethanol stress and heat stress conditions

To provide an orthogonal line of evidence for *HSR* gene interaction, we employed fluorescence microscopy to image live cells bearing *LacO*-tagged *HSP104* and *TetO*-tagged *TMA10* loci in cells expressing GFP-LacI and TetR-mCherry fusion proteins. Both genes are located on Chromosome XII, on opposite arms, and are physically separated by the nucleolus (100–200 rDNA repeats) that lies between them (*Duan et al., 2010*) (schematically depicted in *Figure 7—figure supplement 1A*). In the absence of stress, fluorescence signals representing these two genes are typically well-separated (*Figure 7—figure supplement 1B*, 0 min). Upon heat shock, they rapidly converge, usually within 2.5 min. Upon exposure to ethanol, gene convergence is also observed, albeit less rapidly (*Figure 7—figure supplement 1B*; see also below). Despite the slight delay, these results demonstrate that *HSP104* and *TMA10* coalesce in ethanol stressed cells and do so with similar frequency as in thermally stressed cells (*Figure 7—figure supplement 1C*), consistent with the 3C analysis described above.

Having confirmed the physical interaction of Hsf1-dependent genes under ethanol stress, we assessed the transcriptional status of coalesced genes. Our RT-qPCR analysis indicated that the increase in *HSR* mRNA levels in ES-induced cells is delayed compared to HS-induced cells (*Figure 3*). To obtain insight into *HSR* gene transcription kinetics in single cells, we integrated a stem loop array (24xMS2) upstream of *HSP104*, allowing production of a chimeric transcript visualized upon binding of the MCP-mCherry fusion protein (*Haim et al., 2007*). This strain also harbored *LacO*-tagged *HSP104* and *HSP12* genes and expressed GFP-LacI (schematically illustrated in *Figure 7A*). We were unable to detect an MCP-mCherry focus adjacent to *HSP104* under no stress conditions (*Figure 7B*; 0 min), consistent with very low *HSP104* basal transcript levels (*Figure 3A*). Heat shock induced rapid coalescence between *HSP104* and *HSP12*, as well as transcription from *HSP104*. These phenomena were detectable by 2.5 min as a merged signal of the chimeric transcript and the two GFP-labeled genes (*Figure 7B*). This visualization method allowed us to quantify the percentage of the population that is actively engaged in transcription, revealing that during heat shock, transcription, and coalescence are positively correlated (*Figure 7C and F*).

A detailed live cell analysis supports the strong spatiotemporal correlation between *HSR* gene coalescence and transcription in heat-shocked cells (*Figure 7D*). In contrast, under ethanol stress, *HSP104* RNA was not detected in most cases until 10 min even though *HSP12* and *HSP104* coalesced as early as 2.5 min (*Figure 7E*; *Figure 7—figure supplement 1D*). Indeed, such an analysis suggests temporal uncoupling between *HSR* gene coalescence and *HSR* gene transcription in ethanol stressed cells (*Figure 7F*). Underscoring the disconnect between 3D genome repositioning and transcription is the fact that in ethanol stressed cells, enhanced *HSP12* mRNA levels were undetectable until 60 min (*Figure 3B* and *Figure 3—figure supplement 1*). Collectively, our RT-qPCR, 3C and imaging data argue that ethanol stress induces striking topological changes in *HSR* genes within the first 10–20 min, yet these are accompanied by a minimal increase in transcript levels. This provides a sharp contrast to heat-shocked cells where there exists a strong temporal correlation between *HSR* gene transcription and *HSR* gene repositioning (*Figure 7F*).

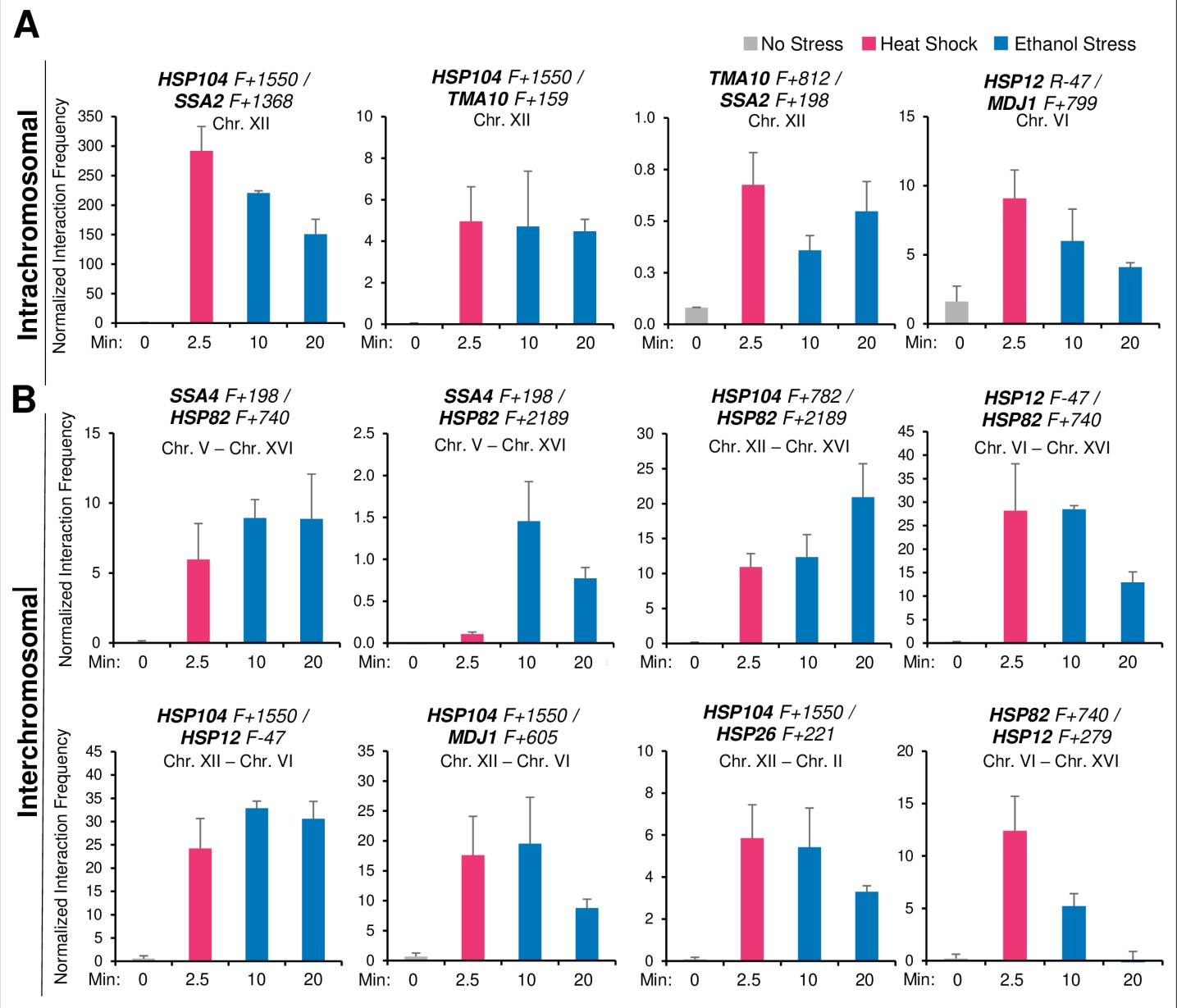

**Figure 5.** Ethanol stress induces intergenic interactions between *HSR* genes that are comparable to those induced by acute thermal stress. (**A**) Intrachromosomal *(cis)* interactions between *HSR* genes were analyzed by Taq I-3C. W303-1B cells were instantaneously shifted from 30° to 39°C for 2.5 min (HS) or exposed to 8.5% v/v ethanol at 30°C for 10 or 20 min (ES). No stress samples were kept at 30°C. Location of Taq I coordinates are provided in *Figure 5—figure supplement 1*. F (forward) primers are positioned near the indicated Taq I restriction site. 3C signals were normalized to the 3C signal derived from using a naked genomic DNA template. Graphs depict means + SD; N=2; qPCR=4. (**B**) Interchromosomal *(trans)* interactions between *HSR* genes were detected as in (**A**).

The online version of this article includes the following source data and figure supplement(s) for figure 5:

**Source data 1.** Spreadsheet tabulates 3C data of cells under no stress, 2.5 min heat shock, and 10- or 20 min ethanol stress.Intrachromosomal and interchromosomal interactions are analyzed (*Figure 5A and B*).

**Figure supplement 1.** Physical maps of genes used in this study.

**Figure supplement 2.** Intragenic interaction frequency induced by ethanol stress is comparable to that induced by thermal stress, despite modest transcriptional output.

**Figure supplement 2—source data 1.** Spreadsheet tabulates intragenic interactions in *HSP104* under HS and ES conditions, compared to *HSP104* mRNA levels under the same conditions (*Figure 5—figure supplement 2A*).

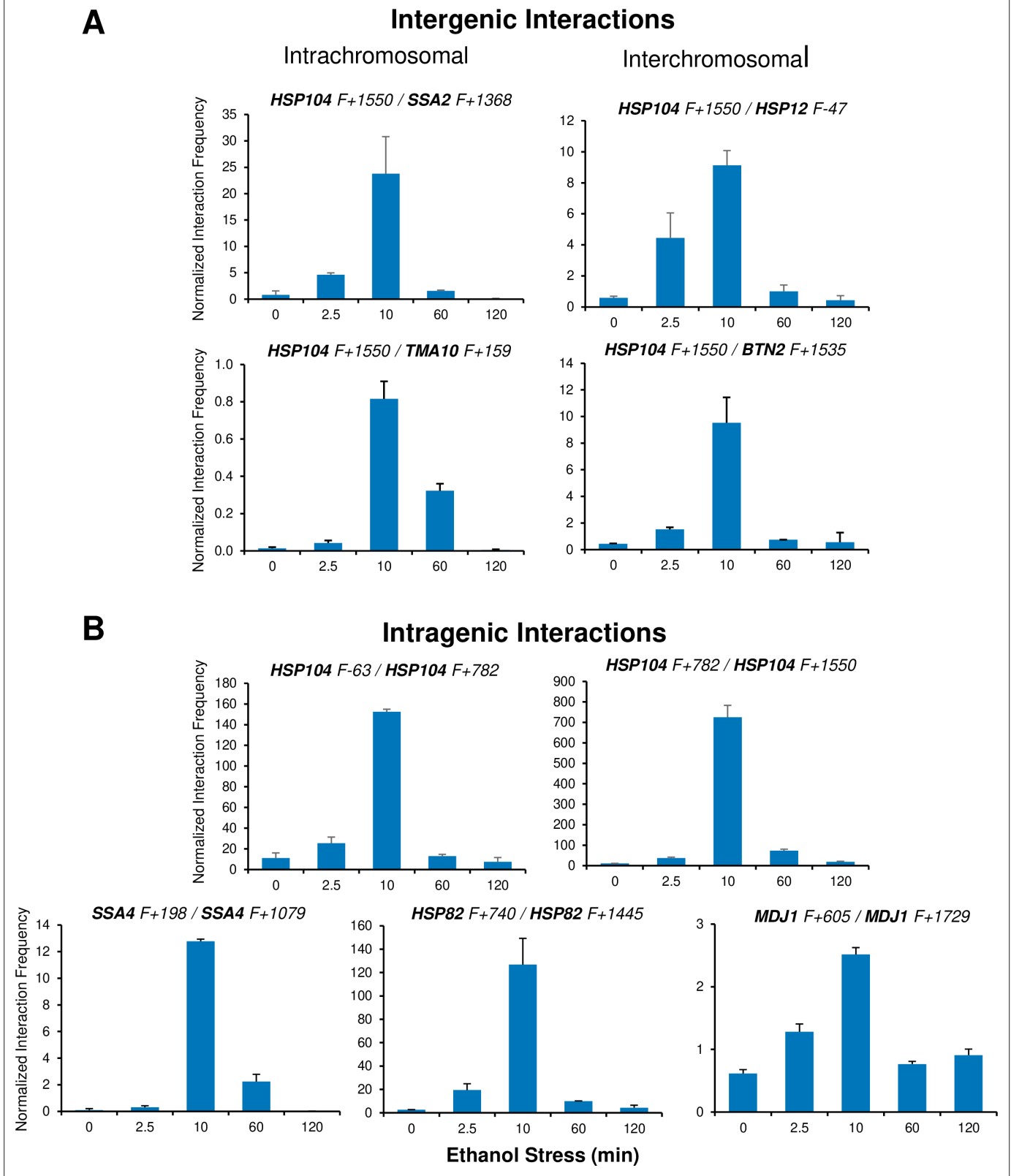

**Figure 6.** Ethanol-induced *HSR* gene interactions are detectable by 2.5 min but typically dissipate within 60 min. (**A**) Taq I-3C analysis of intergenic interactions occurring during ethanol stress was conducted as described in *Figure 5*. All samples were kept at 25°C. Plotted are means + SD. N=2, qPCR=4. (**B**) As in (**A**), but for intragenic interactions.

*Figure 6 continued on next page*

*Figure 6 continued*

The online version of this article includes the following source data for figure 6:

**Source data 1.** Spreadsheet tabulates 3C data of cells subjected to chronic ethanol stress (*Figure 6*).

## Ethanol stress induces formation of long-lived Hsf1 condensates

Recently, TF condensates have been proposed as a mechanism for transcriptional regulation (*Boija et al., 2018*; *Cho et al., 2018*; *Hnisz et al., 2017*; *Nair et al., 2019*; *Sabari et al., 2018*). Heat shock-activated Hsf1 forms small nuclear condensates (diameter of ≤300 nm) that localize at *HSR* gene loci in both human (*Zhang et al., 2022*) and budding yeast cells (*Chowdhary et al., 2022*), and their presence positively correlates with *HSR* gene transcriptional activity. The tendency of Hsf1 to form condensates may be linked to its extensive intrinsically disordered structure (*Figure 8A*), a feature proposed to be critical for the biomolecular condensation of proteins (reviewed in *Alberti et al., 2019*; *Banani et al., 2017*). Additionally, nuclear stress bodies comprised of HSF1 bound to satellite III DNA repeats in human cells have been described (*Jolly et al., 1997*; *Jolly et al., 2002*), although formation of these large foci (diameter of several μm) is independent of the sites of active *Heat Shock Response* genes and their presence may repress *HSR* gene activation (*Gaglia et al., 2020*).

Given the temporal link between Hsf1 condensation and transcription established in heat-shocked yeast cells, we anticipated that the appearance of Hsf1 condensates in ethanol-stressed cells would be delayed relative to what is seen in heat-shocked cells. However, we found that ethanol stress induced formation of Hsf1-GFP condensates as rapidly as did heat shock. These were visible in virtually all ES cells as early as 2.5 min, paralleling their rapid appearance in HS cells (*Figure 8B–D*). However, in contrast to the rapid dissolution of condensates in HS cells, those formed in cells exposed to 8.5% ethanol showed no evidence of dissipating even after 60 min of continuous exposure (*Figure 8B and D*). An independent analysis of Hsf1 tagged with a monomeric GFP, mNeonGreen, gave virtually identical results:>90% of cells exhibited Hsf1 puncta as early as 2.5 min and such puncta were stably maintained for 150 min (*Figure 8—figure supplement 1*). Moreover, in an independent analysis they remained visible at 5.5 hr (data not shown). Therefore, in 8.5% ethanol-stressed cells, formation of Hsf1 condensates is uncoupled from *HSR* gene transcription and their maintenance is uncoupled from *HSR* gene repositioning. An important implication is that although condensates may initiate or promote *HSR* gene repositioning, they cannot maintain the 3D restructured state of the genome.

Does the enhanced stability of ES-induced puncta arise from the nature of the stress or its intensity? To test this, we examined the behavior of yeast cells exposed to a lower concentration of ethanol. As shown in *Figure 8—figure supplement 1*, Hsf1 puncta formed in >95% of 5% ethanol-treated cells within 2.5 min, resembling what was seen in cells exposed to either 8.5% ethanol or 39°C HS. Yet such puncta dissipated in a majority of 5% ES cells within 30 min, resembling the case with HS but strongly contrasting to what was seen with 8.5% ethanol. Therefore, although 5% and 8.5% ES elicit a similar pattern of Sis1 subcellular relocalization, one that is distinct from that caused by either a 39° or 42°C HS (*Figure 2*), it is the greater intensity of 8.5% ethanol stress, not its intrinsic nature, that most closely correlates with the stability of Hsf1 condensates.

## Hsf1 and Pol II are required for *HSR* gene interactions in response to both heat shock and ethanol stress

The above analyses reveal several unexpected differences in the way yeast responds to ethanol stress versus heat stress. Given these differences, we asked whether either Hsf1 or RNA Pol II are required for the repositioning of *HSR* genes in response to ethanol stress; both have been shown to be necessary for 3D genome restructuring in response to heat shock (*Chowdhary et al., 2019*; *Chowdhary et al., 2022*). To do so, we used the auxin-induced degradation system to conditionally degrade Hsf1 and the Rpb1 subunit of RNA Pol II in appropriately engineered strains. As schematically summarized in *Figure 9A*, cells expressing degron-tagged Hsf1 or Rpb1 were pre-treated with 1 mM indole-3-acetic acid (IAA) for 30–40 min, at which time each protein was >90% degraded (*Figure 9—figure supplement 1A*). Although minimal short-term growth defects were detectable, loss of Rpb1 and Hsf1 resulted in loss of cell viability at all temperatures (*Figure 9—figure supplement 1B, C*).

Consistent with previous observations (*Chowdhary et al., 2019*), intergenic interactions between *HSR* genes, located on the same or different chromosomes, were nearly obviated in cells conditionally

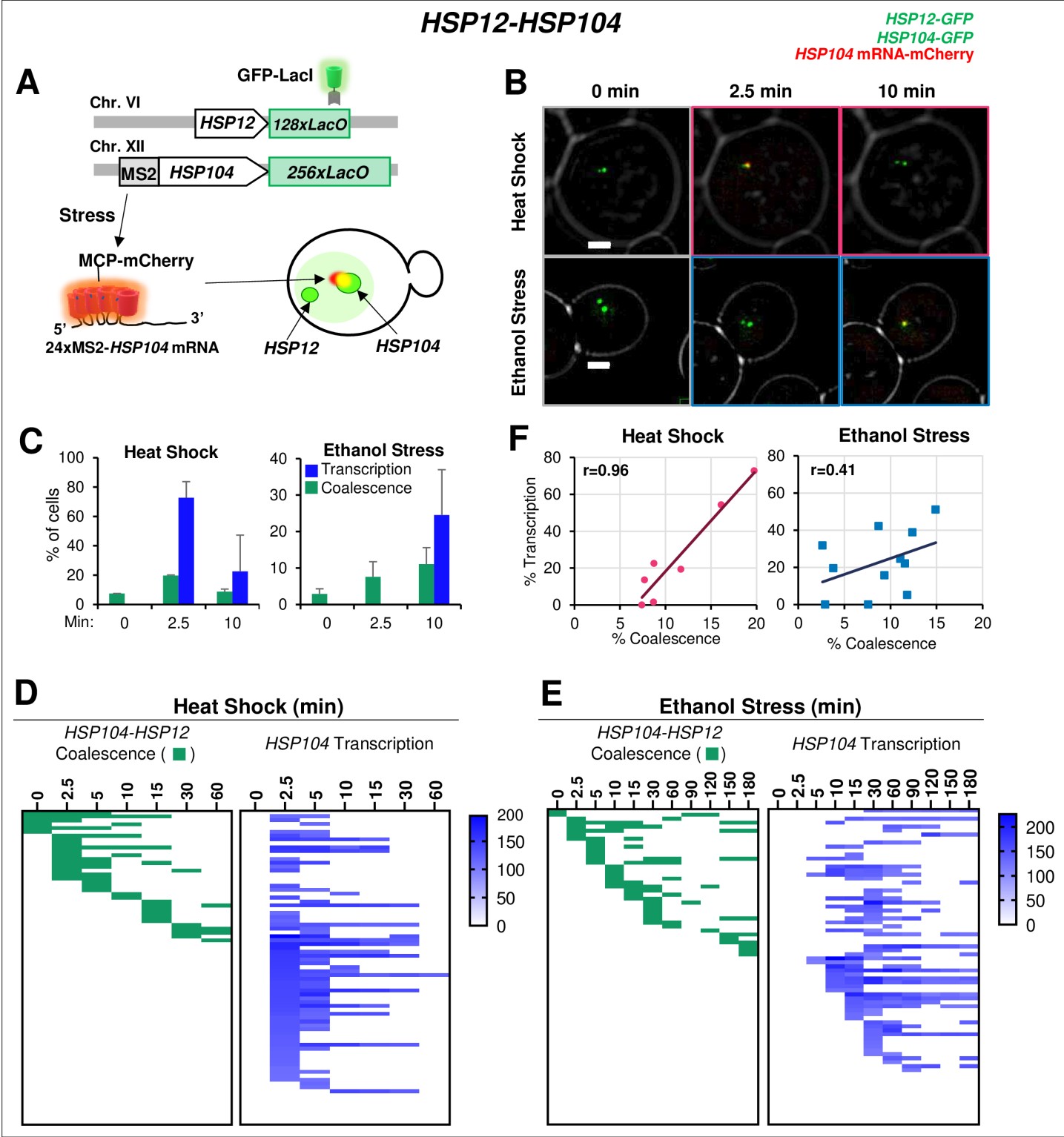

**Figure 7.** *HSR* gene transcription and coalescence are strongly correlated in heat-shocked but not ethanol-stressed cells. (**A**) *HSP12* and *HSP104* are flanked by *LacO* arrays in the heterozygous diploid strain VPY705. In addition, *HSP104* has a 24xMS2 loop array integrated within its 5'-UTR. MCP-mCherry binds to the nascent chimeric *HSP104* transcript and is visualized as a red dot adjacent to the gene which appears as a green dot. (**B**) Live cell confocal fluorescence microscopy of strain VPY705 heat-shocked at 39°C using a VAHEAT device or exposed to 8.5% ethanol at 25°C for the indicated times. An Olympus spinning disk confocal microscope system was used for imaging. Scale bar: 2 μm. (**C**) Quantification of VPY705 cells treated as above and scored for the coalescence of *HSP104-HSP12* and the presence of chimeric *MS2x24-HSP104* mRNA. Cells were scored positive for coalescence

*Figure 7 continued*

only when a single green dot could be visualized in the nucleus across the 11 z-planes. Transcription was scored as positive only when a red dot above background could be seen near the large green dot (*HSP104*). Approximately 40 cells were scored per timepoint, per condition. Graphs represent means + SD. N=2. (**D**) Single cell analysis of *HSP12-HSP104* coalescence (green) and *HSP104* transcription (blue) at discrete timepoints over a heat shock time course. Each row in the transcription analysis corresponds to the same cell in the coalescence analysis. Blue gradient represents the intensity of mCherry signal (*HSP104* transcript) in each cell, as quantified by ImageJ/Fiji (v. 1.54f). (**E**) As in (**D**) but for ethanol stress. (**F**) Pearson correlation coefficient analysis showing the correlation (**r**) between percent of cells positive for transcription and percent of cells positive for coalescence under HS and ES conditions (derived from *Figure 7—figure supplement 1D*). Each plotted value corresponds to a different stress timepoint.

The online version of this article includes the following source data and figure supplement(s) for figure 7:

**Source data 1.** Spreadsheet tabulates the percentage of cells with *HSP104-HSP12* coalescence and *HSP104* mRNA foci upon ES or HS treatment (*Figure 7C and F*).

**Figure supplement 1.** *HSR* gene coalescence and transcription are temporally uncoupled in ethanol stressed cells.

**Figure supplement 1—source data 1.** Spreadsheet tabulates the percentage of cells with *HSP104-TMA10* coalescence under HS or ES conditions (*Figure 7—figure supplement 1C*).

depleted of either Hsf1 or Rpb1 and then heat shocked (*Figure 9B* and data not shown). While a similar strong dependence on Hsf1 is observed in 8.5% ethanol-treated cells, residual *HSR-HSR* gene interactions are retained in Rpb1-depleted cells for certain loci. Particularly noteworthy is the inducible interaction between *HSP12* and *HSP26*, genes co-regulated by Msn2 and Hsf1. Neither is detectably activated following a 10 min exposure to ethanol (*Figure 3B*), yet the genes engage in an intergenic interaction that is unaffected by prior degradation of Rpb1 (*Figure 9B*). Together with the kinetic uncoupling of *HSR* gene repositioning with Pol II recruitment/transcription and the relative permanence of Hsf1 condensates described above, these observations raise the possibility that 8.5% (v/v) ethanol-induced Hsf1 condensates are compositionally different from those formed in response to heat shock and drive *HSR* gene repositioning and transcription in a mechanistically distinct way (see *Figure 10* for model).

## Discussion
### Ethanol stress induces *HSR* gene transcription, *HSR* gene coalescence and formation of Hsf1 condensates

Here we have shown that exposure of budding yeast to a high, but sub-lethal, concentration of ethanol strongly stimulates the binding of Hsf1 to the upstream regulatory regions of *HSR* genes. Unexpectedly, such binding – which is evident as early as 2.5 min – does not lead to concurrent recruitment of Pol II and transcription of *HSR* genes. Instead, Pol II recruitment and transcription are delayed, typically for 10 min or longer. As exposure to 8.5% ethanol causes a global yet transient increase in H3 occupancy – which we interpret as an increase in the compaction of chromatin (see below) – the increase in nucleosome density may present a barrier to both Pol II recruitment and elongation. In addition, another feature of heat shocked-induced Hsf1 activation, repositioning of *HSR* genes within the 3D genome, is observed in cells exposed to ethanol. However, unlike transcription, this phenomenon occurs rapidly and is transient, resembling what is observed in heat shocked cells. The lack of temporal linkage between *HSR* gene transcription and *HSR* intergenic interactions is consistent with the idea that *HSR* gene coalescence and transcription are distinct phenomena and that Hsf1 can instigate long-range changes in 3D genome structure independently of inducing transcription. Thus, Hsf1 represents an example of a gene-specific TF that has functions independent of regulating transcription (discussed further below).

It has recently been demonstrated that in response to heat shock, inducible transcriptional condensates drive 3D genome reorganization in budding yeast. This conclusion arose from several features, including the tight temporal linkage between Hsf1 condensation and *HSR* intergenic interactions and the similar sensitivity of these two phenomena to the aliphatic alcohol, 1,6-hexanediol (*Chowdhary et al., 2022*). Similar to previous observations of heat-shocked cells (and confirmed here), we have found that in cells exposed to 8.5% ethanol, Hsf1 forms discrete puncta within 2.5 min. However, Hsf1-containing condensates formed in response to 8.5% ES are stable – persisting for hours – while such assemblies dissipate within 30 min in cells exposed to HS. In addition, while Hsf1 condensates

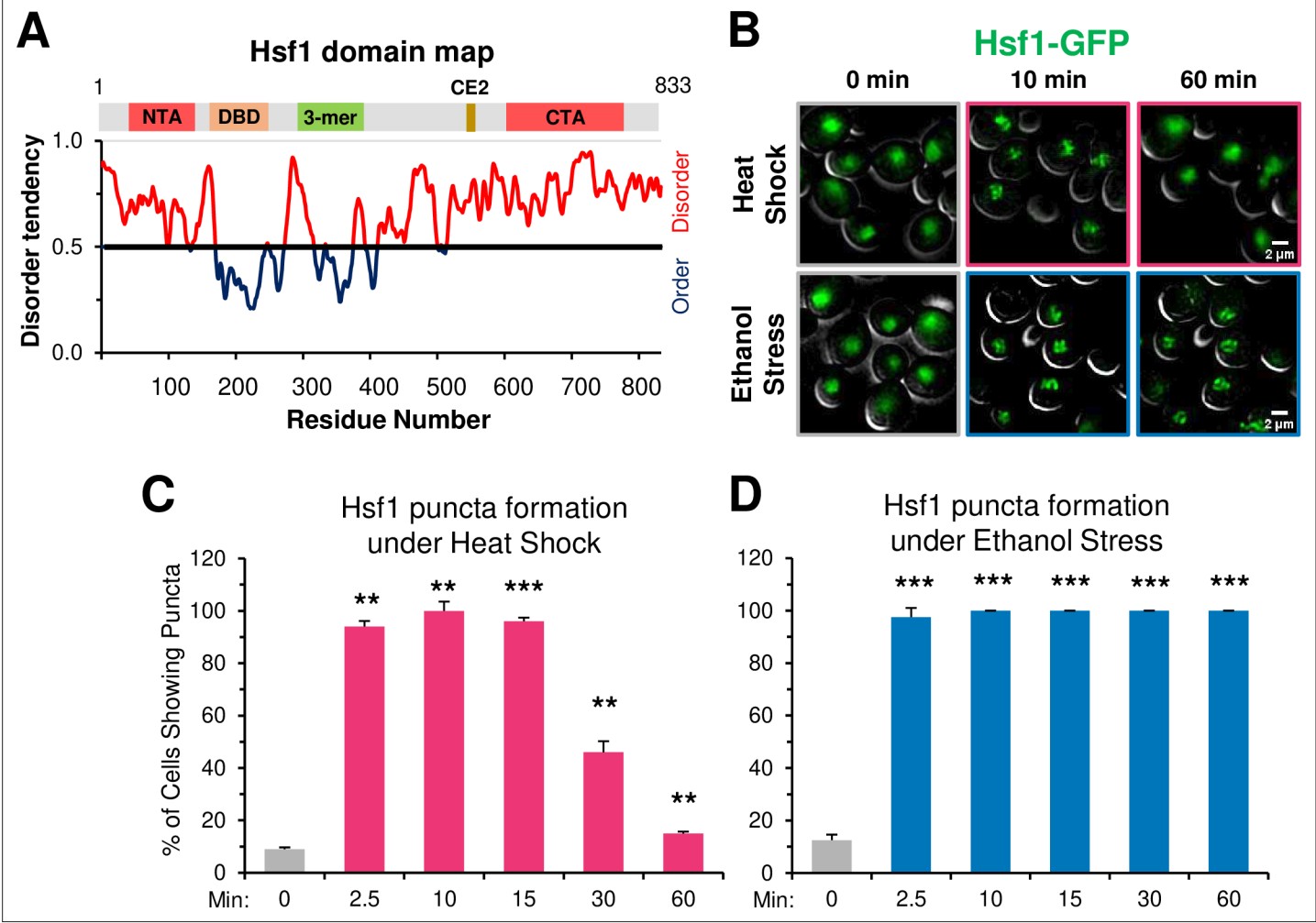

**Figure 8.** Ethanol stress induces rapid formation of long-lasting Hsf1 condensates. (**A**) Hsf1 is a transcription factor bearing N- and C-terminal domains with high disorder tendency (determined by IUPRED2). NTA, N-terminal activator; DBD, DNA binding domain; 3-mer, trimerization domain; CE2, conserved element 2 (Hsp70 binding site); CTA, C-terminal activator. (**B**) Hsf1-GFP condensates form in response to ethanol stress. Diploid cells expressing Hsf1-GFP (ASK741) were grown in synthetic complete medium supplemented with adenine (SDC +Ade) and mounted onto ConA-coated coverslips. Live cell widefield microscopy was performed on cells exposed to either heat shock (38°C) or ethanol stress (8.5% v/v) or left untreated (25°C). A representative plane is shown for each condition out of 11 z-planes imaged (interplanar distance of 0.5 µm). Scale bar: 2 µm. (**C**) ASK741 cells were subjected to a 38°C heat shock for the indicated times and scored for the presence of Hsf1 condensates. A cell was scored as positive if it contained at least one clearly defined puncta. Approximately 200 cells were evaluated per timepoint. A one-tailed t-test was used to assess significance (stress versus no stress condition). N=2. **, p<0.01; ***, p<0.001.(**D**) ASK741 cells were exposed to 8.5% v/v ethanol for the indicated times and the presence of Hsf1 condensates were scored from a total of 200 cells per timepoint. Significance was determined as in (**C**).

The online version of this article includes the following source data and figure supplement(s) for figure 8:

**Source data 1.** Spreadsheet tabulates the percentage of cells containing Hsf1-GFP puncta under heat shock or ethanol stress (*Figure 8C and D*).

**Figure supplement 1.** Hsf1-mNeonGreen rapidly forms condensates in response to both thermal and ethanol stress.

**Figure supplement 1—source data 1.** Spreadsheet tabulates the percentage of cells containing Hsf1-mNeonGreen puncta under heat shock, 8.5% v/v ethanol stress or 5% v/v ethanol stress (*Figure 8—figure supplement 1B*).

formed equally rapidly in cells exposed to 5% and 8.5% ethanol, those formed in the presence of 5% ethanol proved to be transient, resembling their counterparts in HS cells. Relevant to this observation is the finding that Hsf1 activation in yeast exposed to 5% ethanol is dependent on the misfolding of nascent proteins, while in cells exposed to 8% ethanol, Hsf1 activation is dependent on the misfolding of both nascent and mature proteins (*Tye and Churchman, 2021*). Therefore, the persistence of Hsf1 condensates in 8.5% ethanol may derive from widespread unfolded mature proteins that are more difficult to refold than short nascent polypeptides attached to ribosomes.

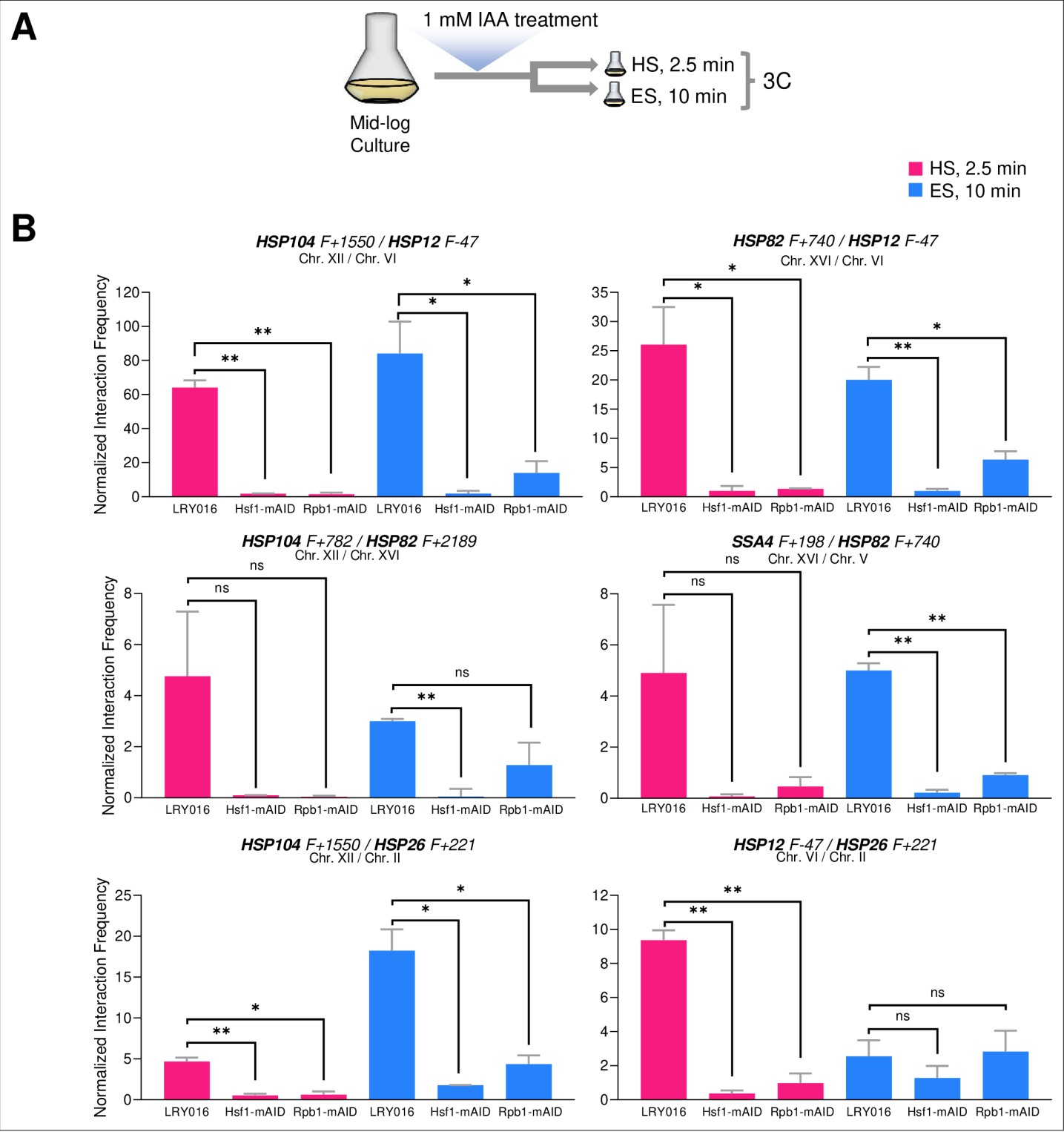

**Figure 9.** Hsf1 and Pol II are critically required for *HSR* gene interactions in response to either heat shock or ethanol stress. (**A**) Experimental strategy. Degron-tagged cells were treated with 1 mM IAA at 25°C for 30–40 min prior to exposure to either heat shock (HS, 39°C for 2.5 min) or ethanol stress (ES, 8.5% v/v ethanol for 10 min) followed by HCHO crosslinking and 3C analysis. (**B**) Strains LRY016 (W303-1B; OsTIR1), LRY100 (LRY016; Hsf1-mAID), and LRY102 (LRY016; Rpb1-mAID) were subjected to the above protocol and physical interactions between the indicated chromosomal loci were detected by Taq I-3C as in *Figure 5*. Representative interchromosomal interactions are shown. Graphs represent means + SD. Statistical significance between the indicated interaction frequencies was determined using a one-tailed t-test. *, *p*<0.05; **, *p*<0.01; *ns*, not significant. A no stress sample,

*Figure 9 continued on next page*

*Figure 9 continued*

maintained at 30°C for 10 min following IAA treatment and then crosslinked, was handled in parallel. No signal above background was detected for any pairwise test.

The online version of this article includes the following source data and figure supplement(s) for figure 9:

**Source data 1.** Spreadsheet tabulates 3C analysis of cells pretreated with 1 mM auxin followed by 2.5 min HS or 10 min ES (*Figure 9B*).

**Figure supplement 1.** Hsf1 and Rpb1 are efficiently degraded in degron-tagged strains following addition of auxin.

**Figure supplement 1—source data 1.** Raw files for immunoblots of Hsf1-mAID-9xMyc, Rpb1-mAID-9xMyc and the corresponding Pgk1 as loading control in cells treated with 1 mM auxin at 25°C (*Figure 9—figure supplement 1A*).

**Figure supplement 1—source data 2.** Labeled blots for Hsf1-mAID-9xMyc, Rpb1-mAID-9xMyc, and the corresponding Pgk1 as loading control in cells treated with 1 mM auxin (IAA) at 25°C (*Figure 9—figure supplement 1A*).

**Figure supplement 1—source data 3.** Spreadsheet tabulates relative Hsf1-mAID-9xMyc and Rpb1-mAID-9xMyc protein levels, normalized to Pgk1 and 0 min of treatment with 1 mM auxin (*Figure 9—figure supplement 1A*).

Additional features might account for the way in which these two proteotoxic stresses elicit the HSR in budding yeast. For example, while ethanol denatures proteins similar to thermal stress (*Furutani and Izawa, 2022*; *Kato et al., 2011*), it impacts the cell at other levels. Actively growing yeast preferentially consume sugars and even in the presence of oxygen can produce ethanol (*Dashko et al., 2014*). Once glucose is depleted, cells start consuming ethanol as an alternative carbon source. This change in carbon source, termed the diauxic shift, is marked by a reduction in growth rate due to the less efficient generation of energy as well as accumulation of waste products and depletion of nutrients (*Herman, 2002*). Consistent with activation of the HSR, this physiological adjustment is accompanied by increased chaperone synthesis (*Piper, 1995*; *Piper et al., 1994*) as well as an increase in the activity of alcohol metabolism-related enzymes. Such an adaptation likely takes time. In contrast to the gradual increase in ethanol that occurs under natural conditions, here we have imposed an instantaneous ethanol shock. The addition of ethanol elicits the heat shock transcriptional response and synthesis of chaperones. The cell growth arrest seen here (*Figure 1A*) may arise as part of the adaptive program deployed upon encountering ethanol stress, such as a transition into quiescence. However, we fail to see activation of Msn2 targets, a signature of quiescence (*Breeden and Tsukiyama, 2022*), early in the exposure to ES at a time when Hsf1 itself is activated (*Figure 3B*). While other possibilities exist, our results suggest that the activation of programs involved in proteome protection are likely the first line of defense against sudden ethanol stress. Salient differences by which yeast cells respond to heat versus ethanol stress are summarized in *Table 1*; a graphical summary highlighting such differences is presented in *Figure 10A*. A model of the Hsf1 transcriptional response suggested by the data presented here and elsewhere is provided in *Figure 10B*.

## Exposure to ethanol transiently induces the global compaction of chromatin

An important observation is that exposure of cells to 8.5% ethanol leads to a widespread increase in H3 ChIP signal that we interpret as increased compaction of chromatin. Although this effect is temporary, both euchromatic and heterochromatic regions are impacted, and this is consistent with measurements of total chromatin volume that reveal a decrease lasting nearly 60 min. A similar outcome was recently reported for both yeast and mammalian cells exposed to aliphatic di-alcohols (*Itoh et al., 2021*; *Meduri et al., 2022*). It is probable that the effect of ethanol exposure is nearly instantaneous since, as discussed above, ethanol is known to denature proteins (*Kato et al., 2019*), likely by dehydration (disruption of biomolecules' hydration shell). While such denaturation may contribute to chromatin compaction, the effect is reversible, possibly due to refolding / renaturation mediated by Hsp70 and other molecular chaperones whose intracellular concentration increases during ethanol exposure. As mentioned above, this temporary increased density of chromatin could suppress Pol II recruitment and subsequent elongation. Also not ruled out is a direct effect of 8.5% ethanol on one or more components of the Pol II machinery. Possible advantages of chromatin compaction could be to downregulate global transcription, as well as to limit chromatin damage by reactive oxygen species or other potentially damaging molecules present in the cell during ethanol stress (*Bradley et al., 2021*; *Costa et al., 1997*; *Davidson et al., 1996*; *Davidson and Schiestl, 2001*; *Shen et al., 2020*; *Voordeckers et al., 2020*).

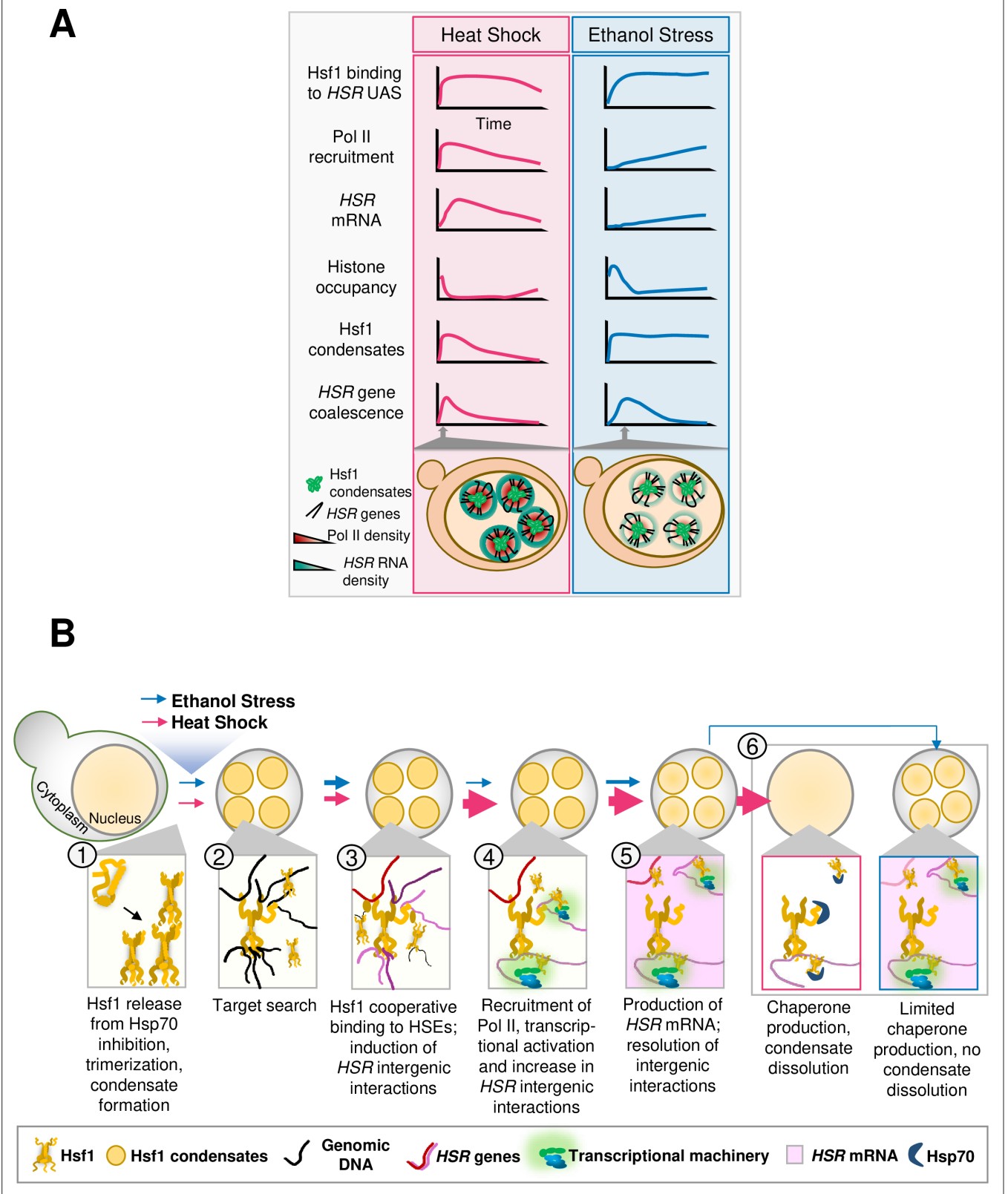

**Figure 10.** Central findings of this study. (**A**) Kinetics of the yeast HSR to thermal and chemical stresses. (**B**) Hsf1 forms condensates, restructures the genome and transcriptionally activates *HSR* genes in a distinct fashion in response to thermal vs. chemical stress. *Heat Shock (pink arrows): S. cerevisiae* cells exposed to heat shock (30° to 39°C upshift) undergo protein misfolding (proteotoxicity), leading to chaperone titration and activation of Hsf1 accompanied by formation of Hsf1 condensates (1). Rapid target search is hypothesized to occur via multivalent interactions between Hsf1's activation

*Figure 10 continued on next page*

*Figure 10 continued*

domains and chromatin-bound proteins (*Brodsky et al., 2020*) (2). Subsequently, Hsf1 cooperatively binds HSEs and induces the 3D reorganization of *Heat Shock Responsive (HSR)* genes (3). Simultaneously, RNA Pol II is recruited to *HSR* genes and transcription is induced (4). Productive transcription (pink shading) ensues, followed by dissolution of *HSR* gene interactions (5). Rapid export of *HSR* mRNAs (*Zander et al., 2016*) facilitates production of chaperone and cytoprotective proteins, which aid in restoration of proteostasis and disassembly of Hsf1 condensates (*Chowdhary et al., 2022*) (6). Thickness of arrows (for both HS and ES) symbolizes rapidity and/or magnitude of the subsequent step. *Ethanol Stress (blue arrows):* Exposure to 8.5% (v/v) ethanol induces proteotoxicity, titration of chaperones, and subsequent activation of Hsf1, triggering formation of condensates (1, 2). Condensates may aid Hsf1 target search by facilitating the multivalent interactions described above (2). Hsf1 then binds HSEs, inducing *HSR* gene repositioning (3). Subsequently, Pol II is recruited and transcription is induced (4). Ethanol stress-induced interactions dissipate well before maximal *HSR* gene transcriptional induction is achieved (5). The weak transcriptional *HSR* gene output, coupled with suppressed *HSR* mRNA export under ES (*Izawa et al., 2008*), results in a low level of chaperone synthesis (6) that fails to resolve proteotoxicity. This likely contributes to the persistence of Hsf1 condensates.

## A novel function for a TF that is uncoupled from regulating the transcription of its target gene

A key finding is that in response to ethanol stress, *HSR* genes reposition and Hsf1 condensates form well before transcription of Hsf1-dependent genes peaks, and in certain cases, is even detected. This suggests that Hsf1 has a function distinct from transactivating its target genes: it drives *HSR* gene repositioning that culminates in their physical coalescence. Likewise, deletion of the N-terminal IDR (NTA, amino acids 2–146; see *Figure 8A*) was observed to have little effect on *HSR* gene transcription during an acute heat shock yet formation of Hsf1 condensates and intergenic 3C interactions were suppressed (*Chowdhary et al., 2022*). Furthermore, as discussed above, recruitment of Hsf1 to its target HSEs is delayed in response to ES versus HS. Nevertheless, formation of Hsf1 condensates occurs with similar kinetics under both conditions, well prior to the peak of Hsf1 DNA binding in ES cells. This suggests the possibility that two types of Hsf1 clusters exist: one that is DNA-bound (detected by ChIP) and the other that hovers over the *HSR* genes (detected by imaging). It is possible that this cloud-forming Hsf1 cluster aids in target searching, by analogy with Gal4 that can also form clusters without binding DNA (*Meeussen et al., 2023*).

## Conclusion

While Hsf1 is known to induce interactions between transcriptionally active *HSR* genes in response to heat shock, here we have demonstrated a similar role for this transcription factor in response to ethanol stress. Despite minimal *HSR* gene transcription during the initial 10 min exposure to ethanol, *HSR* genes engage in robust physical interactions, rivaling those seen for 2.5 min HS. This interaction correlates with an increase in definition of Hsf1 condensates yet is not accompanied by concurrent recruitment of Pol II to promoters, perhaps due to the compaction of chromatin in ethanol-stressed cells. Hsf1 therefore forms condensates and drives 3D repositioning of its target genes without appreciably activating these genes. Furthermore, Hsf1 does this through formation of condensates that may be materially different than those that form in response to HS. Our results thus argue that Hsf1

**Table 1.** Kinetics of select nuclear phenomena in response to heat shock and ethanol stress.

| Parameter | Heat Shock (39°C) | Ethanol Stress (8.5% v/v) |
| --- | --- | --- |
| Hsf1 binding to HSEs | Rapid, peak at 2.5 min | Slight delay, peak at 10 min |
| Pol II recruitment to *HSR* genes | Rapid, peak at 2.5 min | Severely delayed, detection starts at 10 min |
| Histone H3 occupancy at *HSR* genes | Rapid depletion | Transient increase, followed by gradual depletion |
| Genome compaction* | Transient | Sustained |
| *HSR* gene transcription | Rapid | Delayed |
| *HSR* gene coalescence | Rapid, transient | Slightly delayed, transient |
| Hsf1 condensates | Rapidly induced, transient, well defined | Rapidly induced, stable, poorly defined |

*As inferred from enhanced H3 ChIP signal at non-Hsf1 regulated loci.

condensate formation, while sufficient for TF DNA binding and 3D genome restructuring, is not sufficient to drive transcription. Additional factors and/or activities – such as the opening of chromatin – are necessary. Further research into the biophysical properties, molecular regulation, and functional consequences of HSF1 condensates will deepen our understanding of how cells respond to both thermal and chemical stress and maintain cellular homeostasis.

# Materials and methods

## Key resources table

| Reagent type (species) or resource | Designation | Source or reference | Identifiers | Additional information |
|---|---|---|---|---|
| Cell line (*S. cerevisiae*) | BY4741 | Research Genetics | | |
| Cell line (*S. cerevisiae*) | W303-1B | Rodney Rothstein | | |
| Recombinant DNA reagent | pFA6a-link-ymNeonGreen-SpHis5 plasmid | Addgene | Cat# 125704 RRID:Addgene_125704 | PMID:30783202 |
| Recombinant DNA reagent | pGZ154 plasmid | C.K. Govind, University of Oakland | | |
| Recombinant DNA reagent | pHyg-AID*–9myc | Addgene | Cat# 99518 RRID:Addgene_99518 | PMID:23836714 |
| Chemical compound, drug | Indole-3-acetic acid (IAA) | Sigma-Aldrich | Cat# I3750 | |
| Chemical compound, drug | Sodium azide | Sigma-Aldrich | Cat# S2002 | |
| Chemical compound, drug | Ethanol | Decon Labs | Cat# 04-355-223 | |
| Chemical compound, drug | Phenylmethylsulfonyl fluoride (PMSF) | Sigma-Aldrich | Cat# P7626 | |
| chemical compound, drug | Phenol:Chloroform:Isoamyl Alcohol mixture | Sigma-Aldrich | Cat# 77617 | |
| Chemical compound, drug | Formaldehyde | Fisher Scientific | Cat# 14-650-250 | |
| Chemical compound, drug | Glycine | Bio-Rad | Cat# 1610724 | |
| chemical compound, drug | Protein A-Sepharose beads | GE Healthcare (Cytiva) | Cat# 17096303 | |
| Commercial assay or kit | High-Capacity cDNA Reverse Transcription Kit | Applied Biosystems | Cat# 4368814 | |
| Commercial assay or kit | iTaq Universal SYBR Green Supermix | BioRad Laboratories | Cat# 1725125 | |
| Antibody | Anti-Btn2 (rabbit polyclonal) | Bernd Bukau, University of Heidelberg | | WB (1:5000) |
| Antibody | Anti-cMyc (mouse monoclonal) | Santa Cruz Biotechnology | Cat# sc-40 RRID:AB_627268 | WB (1:1000) |
| Antibody | Anti-Histone H3 (rabbit polyclonal) | Abcam | Cat# ab1719 | WB (1:1000) ChIP (1 µL/rxn) |
| Antibody | Anti-Hsf1 (rabbit polyclonal) | PMID:8943356 | | ChIP (1.5 µL/rxn); Gross Lab |
| Antibody | Anti-Hsp104 (rabbit polyclonal) | Enzo Life Sciences | Cat# ADI-SPA-1040-F RRID:AB_11181448 | WB (1:1000) |
| Antibody | Anti-Pgk1 (mouse monoclonal) | ThermoFisher Scientific | Cat# 459250 RRID:AB_2532235 | WB (1:10,000) |
| Antibody | Anti-Rpb1 (rabbit polyclonal) | PMID:16199876 | | ChIP (1.5 µL/rxn); Gross Lab |
| Antibody | Anti-Mouse, Horseradish peroxidase conjugated (goat) | Santa Cruz Biotechnology | Cat# sc-2005 RRID:AB_631736 | WB (1:5000) |
| Peptide, recombinant protein | Concanavalin A | Sigma-Aldrich | Cat# C2010 | |

*Continued on next page*

*Continued*

| Reagent type (species) or resource | Designation | Source or reference | Identifiers | Additional information |
|---|---|---|---|---|
| Peptide, recombinant protein | TaqI restriction enzyme | New England Biolabs | Cat# R0149L | |
| Commercial assay or kit | Quick Ligation Kit | New England Biolabs | Cat# M2200L | |
| Software, algorithm | FIJI/ImageJ | https://imagej.net/software/fiji/ | | |
| Software, algorithm | Imaris | https://imaris.oxinst.com/ | | |
| Software, algorithm | IUPRED2 | https://iupred2a.elte.hu/ | | |
| Other | VAHEAT device | Interherence GmbH | | Temperature controller for live cell imaging |

## Yeast strain construction

The *HSF1-mNeonGreen* (*HSF1-mNG*) diploid strain LRY033 expressing a yeast-optimized version of mNeonGreen, derived from *Branchiostoma lanceolatum* (*Shaner et al., 2013*), and co-expressing Sis1-mKate and Hsp104-BFP (Blue Fluorescent Protein; mTagBFP2) was created as follows. First, DPY1561 (*Feder et al., 2021*) was crossed to W303-1B to create strain LRY031. This diploid was sporulated and a strain homozygous for *HSF1* and retention of one allele each of *SIS-mKate* and *HSP104-mTagBFP2* was obtained after back crossing. The resultant diploid was named LRY032. LRY032 was transformed with a PCR amplicon containing 50 bp of homology sequences flanking the *HSF1* stop codon, targeting the mNG tag to *HSF1* at its C-terminus flanked by the *HIS3* selectable marker. The plasmid template for this amplification was pFA6a-link-ymNeonGreen-SpHis5 (*Botman et al., 2019*). LRY033 is heterozygous for *HSF1-mNG, SIS1-mKate* and *HSP104-mTagBFP2.*

Other strains were created as follows. LRY037 was constructed using the *HSF1*-targeted *mNeon-Green* amplicon to transform strain W303-1B. LRY040 was constructed by transforming LRY037 with an amplicon containing *RPB3-mCherry::hphMX6*, obtained using genomic DNA from strain SCY004 (*Chowdhary et al., 2022*) as template. LRY100 and LRY102 were constructed using LRY016 (*Rubio and Gross, 2023*) as recipient of the mini-degron tag amplified from pHyg-AID*–9Myc (*Morawska and Ulrich, 2013*), targeted to the C-terminus of *HSF1* and *RPB1*, respectively.

A complete list of strains as well as plasmids and primers used in strain construction are listed in *Supplementary file 1a, b, and c.*

## Yeast culture and treatment conditions

Cells were grown at 30°C in YPDA (1% w/v yeast extract, 2% w/v peptone, 2% w/v dextrose and 20 mg/L adenine) to mid-log density ($OD_{600}$=0.6–0.8). For ethanol stress, the cell culture was mixed with an equal volume of YPDA containing 17% v/v ethanol (yielding a final concentration of 8.5%) and incubated at 25°C for different lengths of time as indicated in the figures. For heat shock, the mid-log culture was mixed with an equal volume of 55°C YPDA medium to achieve an instantaneous temperature upshift to 39°C, and the culture was maintained at 39°C for the indicated times. The no stress samples were diluted with an equivalent volume of YPDA and maintained at 25°C. Samples were kept at their respective temperatures using a water bath with constant shaking.

## Cell viability and growth assays
### Cell viability assay

Cells were grown at 30°C in YPDA to $OD_{600}$=0.6 and then diluted to $OD_{600}$=0.4 using an equivalent volume of medium as described above for ethanol stress, heat shock or the no stress control (YPDA at 25°C). Cells were kept under these conditions for 3 hr; during this time aliquots were taken at different timepoints and diluted 1:26,000 for plating onto YPDA. Plates were incubated at 30°C for 3 days then scanned. Colonies were quantified using ImageJ/Fiji (v. 1.53t) (*Schindelin et al., 2012*) - 'Analyze Particles' option. The number of colony-forming units (CFUs) obtained in stress samples were normalized to the no stress samples and expressed as a percentage of the number of CFUs obtained in the no stress sample.

## Growth assay

Cells were grown in liquid culture and subjected to the same treatments as described above. $OD_{600}$ readings of each sample were taken at intervals over 3 hr (see *Figure 1*). The average $OD_{600}$ from two samples was plotted versus time.

## Spot dilution

Spot dilution was conducted as described previously (*Rubio and Gross, 2023*).

## Auxin-induced degradation

Cells expressing the F-box protein osTIR1 in combination with a degron-tagged protein were grown in YPDA medium to mid-log phase and indole-3-acetic acid (IAA) was then added to a final concentration of 1 mM. IAA stocks (10 mg/mL [57 mM]) were prepared fresh in 95% ethanol and filter-sterilized before use. For immunoblot analysis, cells were treated for varying times up to 1 hr prior to metabolic arrest achieved through addition of sodium azide to a final concentration of 20 mM, followed by cell harvesting. 0 min control samples were treated with vehicle alone (1.7% v/v ethanol). For growth curve analysis, samples were kept at 30°C with constant shaking and aliquots were removed at various timepoints to monitor $OD_{600}$. For 3C analysis, cells were similarly grown in YPDA to $OD_{600}$=0.6, then treated with 1 mM IAA for either 30 min (LRY016 and LRY100) or 40 min (LRY102) prior to cell cross-linking and subsequent cell harvesting.

## Reverse transcription-qPCR (RT-qPCR)

RT-qPCR was conducted as previously described using 25 mL cell culture aliquots (*Rubio and Gross, 2023*). Transcription was terminated via addition of sodium azide to a final concentration of 20 mM (*Lee and Garrard, 1991*). PCR primers used are listed in *Supplementary file 1d*.

## Immunoblot analysis

Immunoblotting was performed as previously reported using 20 mL cell culture aliquots (*Rubio and Gross, 2023*).

## Chromatin immunoprecipitation

Chromatin immunoprecipitation was performed as previously described (*Chowdhary et al., 2019*) with modifications. Briefly, cells from a 50 mL mid-log culture were exposed to 8.5% v/v ethanol for 0, 2.5, 10, 20 or 60 min and then fixed using 3.01% formaldehyde (HCHO), resulting in a net concentration of 1%. Glycine was added to 0.363 M to quench excess formaldehyde. We note that HCHO is consumed in a reaction with ethanol. Since 8.5% ethanol equals 1.46 M, acetal formation at 1:2 stoichiometry consumes 0.730 M (2.01%) of HCHO, leaving an effective concentration of 0.363 M (1%). The reaction resulting in formation of an acetal is illustrated in *Figure 11*.

For heat shock, cells from a 50 mL mid-log culture were fixed with 1% (0.363 M) formaldehyde following 39°C upshift for the times indicated (*Figure 4*; *Figure 4—figure supplements 1 and 2*). Glycine was then added at a final concentration of 0.363 M to quench unreacted formaldehyde. Chromatin lysates, prepared as previously described (*Chowdhary et al., 2019*), were incubated using 20% of the lysate with one of the following antibodies: 1.5 µL of anti-Hsf1 (*Erkine et al., 1996*), 1.5 µL anti-Rpb1 antiserum (*Zhao et al., 2005*), or 1 µL of H3 antibody (Abcam, ab1791). Incubation of lysate

**Figure 11.** Acetal formation reaction (*McMurry, 2012*).

with antibody was done for 16 hr at 4°C. Chromatin fragments bound to antibody were captured on Protein A-Sepharose beads (GE Healthcare) for 16 hr at 4°C. Wash, elution, and DNA purification were conducted as described (*Chowdhary et al., 2019*). The ChIP DNA template was quantified by qPCR (7900HT Fast Real Time PCR System, Applied Biosystems). A standard curve was generated using genomic DNA and ChIP DNA quantities were deduced by interpolation. The qPCR signal for each primer combination was normalized relative to the corresponding signal arising from the input DNA. Primers used in ChIP analysis are listed in *Supplementary file 1e*.

## Taq I chromosome conformation capture (Taq I-3C)

TaqI-3C was performed as previously described (*Rubio and Gross, 2023*). A master cell culture was grown at 30°C in YPDA from $OD_{600}$=0.15 to a final $OD_{600}$=0.8. Aliquots of 50 mL were used for each condition. Heat shock and ethanol stress were conducted as described above. Primers for analysis of 3C templates are listed *Supplementary file 1f*.

## Fluorescence microscopy
### Widefield fluorescence microscopy

For *Figure 7—figure supplement 1B, C* and *Figure 8*, cells were grown at 30°C in Synthetic Complete Dextrose (SDC) medium supplemented with 0.1 mg/mL adenine to early log phase. From this culture, 90 µL were removed and cells were immobilized onto a concanavalin A (ConA, Sigma Aldrich, 100 µg/mL in $ddH_2O$)-coated coverslip for 20 min. For ethanol stress, the medium was then removed and replaced by either SDC or SDC +8.5% v/v ethanol. The coverslip was mounted onto a concave microscope slide to reduce media evaporation (2-Well Concavity Slide, Electron Microscopy Sciences). Images were acquired using an AX70 Olympus epifluorescence microscope across 11 z-planes with 0.5 µm of interplanar distance. Filter set 89021 (Chroma Technology) and a Photometrics Prime 95B camera were used to image GFP and mCherry. SlideBook Software version 6.0.15 (Intelligent Imaging Innovations) was used for image capturing and z-axis stepping motor operation (Ludl Electronic products).

For heat shock, no stress (NS) control images were taken using an Olympus Ach 100/1.25-NA objective coupled to a heating device (Bioptechs objective heater system). Heat-shocked sample imaging was done by rotating the objective away from the coverslip, switching the heating system on, and allowing it to reach 38°C before returning the objective to the coverslip for an instantaneous heat shock. The same field of view on the coverslip was imaged for both NS and HS timepoints. For ethanol stress, an Olympus UPlanFl 100/1.4-NA objective was used for image acquisition. Analysis of the images after acquisition was done using ImageJ (v. 1.52).

In the *HSP104-TMA10* coalescence analysis, cells in which the fluorophore-tagged genes had undergone replication (two green or two red fluorescent spots, indicative of late S/G2 phase) were excluded from the analysis. The locations of the two tagged loci were analyzed over 11 different z-planes, encompassing the whole nucleus. Cells were deemed coalescence positive if they displayed colocalized, non-resolvable fluorophore signals with a distance of <0.4 µm between centroids.

### Spinning disk confocal microscopy

For all other imaging, cells were grown as described above and image acquisition was done using an UPlan Apo 100 x/1.50 NA objective in an Olympus Yokogawa CSU W1 Spinning Disk Confocal System coupled to sCMOS cameras (Hamamatsu Fusion) controlled by cellSens Dimension software. A 50 µm pinhole disk was used for imaging in combination with 10% of laser power employed for excitation using 405 nm, 488 nm, and 561 nm lasers. Z-stacks were captured as for widefield fluorescence. For heat shock, cells were attached to a VAHEAT substrate (Interherence GmbH, Germany) (*Icha et al., 2022*) using ConA as above. The substrate was mounted onto the VAHEAT holder and control images were captured before heat was applied. Samples were instantaneously heated to 39° or 42°C. The substrate reservoir was covered with a coverslip to prevent media evaporation. Imaging was done over multiple timepoints as indicated in the figures. For ethanol stress, cells were mounted onto a coverslip as above, petroleum jelly was used to hold the coverslip against the concave slide and to reduce media evaporation. The slide was inverted and placed on the stage for visualization at room temperature.

Image reconstruction and analysis were done using FIJI/ImageJ (v. 1.53t) (*Schindelin et al., 2012*). Hsp104-BFP (LRY033) and Hsf1-mNeonGreen (LRY040) foci count were performed using the 'Cells' feature in Imaris (v.10.0.0). The background fluorescence was used to delimit the region of interest (ROI). This ROI was then used to quantify the number of foci per cell. For Hsp104, the ROI (whole cell) was delimited to volumes between 20 and 140 $\mu m^3$ (*Uchida et al., 2011*). Foci size was measured to an average of 0.4 $\mu m$ in diameter, which was used to train Imaris to recognize foci of distinct sizes ('Different Spot Sizes [Region Growing]' option). A similar approach was taken to identify the nucleus (as the cell) and the Hsf1 foci (as nucleus) using the 'Cells' feature in Imaris. Nuclei with volume of 10 $\mu m^3$ ±3.7 $\mu m^3$ were used for the analysis (*Larson et al., 2011*). Data obtained from these analyses were plotted using GraphPad Prism 8. Plot profile for signal arising from Hsf1-mNeonGreen, Sis1-mKate, and Hsp104-BFP was done using FIJI/ImageJ (v. 1.53t).

For analysis of transcription, *24xMS2-HSP104* mRNA was visualized using the signal arising from MCP-mCherry binding to the chimeric transcript. Cells were interrogated for transcription by assaying the presence of an mCherry focus adjacent to *HSP104-LacO$_{256}$* bound by LacI-GFP. The brightest voxel next to *HSP104* was used as a proxy for the intensity of transcription and plotted for each cell over the different timepoints. Intensity was measured using FIJI/ImageJ (v. 1.53t) and plotted employing GraphPad Prism 8.

### Statistical tests

The statistical significance of the differences in mean values for a variety of assays was determined as specified in the figure legends using GraphPad Prism 8.

### Materials availability statement

Yeast strains, plasmids and antibodies are available upon request. All data generated in this study are presented in the manuscript, source data, and supplementary files.

## Acknowledgements

We thank Drs. Kelly Tatchell, Lucy C Robinson, Rini Ravindran, Eric First, Surabhi Chowdhary and David Pincus for helpful discussions and technical advice; Drs. Vickky Pandit and Amoldeep Kainth for strain construction; Paula Polk for help with quantitative PCR; Drs. Axel Mogk and Bernd Bukau, University of Heidelberg, for the Btn2 antibody; Drs. Chabbi Govind, Oakland University, and Helle D Ulrich, Institute of Molecular Biology gGmbH, for OsTIR1 and mAID plasmids; and Drs. Donna and Jason Brickner, Kelly Tatchell, Surabhi Chowdhary, Amoldeep Kainth and David Pincus for generously providing yeast strains. This work was supported by NIH grants R15 GM128065 and R01 GM138988 awarded to D S G and an Ike Muslow predoctoral fellowship awarded to L S R.

## Additional information

### Funding

| Funder | Grant reference number | Author |
| --- | --- | --- |
| National Institutes of Health | R01 GM138988 | David S Gross |
| National Institutes of Health | R15 GM128065 | David S Gross |
| Ike Muslow Predoctoral Fellowship | | Linda S Rubio |

The funders had no role in study design, data collection and interpretation, or the decision to submit the work for publication.

### Author contributions

Linda S Rubio, Conceptualization, Formal analysis, Funding acquisition, Validation, Investigation, Visualization, Methodology, Writing – original draft, Writing – review and editing; Suman Mohajan, Formal

analysis, Validation, Investigation, Visualization, Methodology; David S Gross, Conceptualization, Resources, Formal analysis, Supervision, Funding acquisition, Investigation, Visualization, Writing – original draft, Project administration, Writing – review and editing

### Author ORCIDs
Linda S Rubio https://orcid.org/0000-0002-7260-6971
Suman Mohajan http://orcid.org/0000-0002-8913-9425
David S Gross https://orcid.org/0000-0002-7957-8790

Reviewer #2 (Public Review): https://doi.org/10.7554/eLife.92464.4.sa1
Reviewer #3 (Public Review): https://doi.org/10.7554/eLife.92464.4.sa2
Author response https://doi.org/10.7554/eLife.92464.4.sa3

## Additional files

### Supplementary files
• Supplementary file 1. Supplemental tables. (a) Yeast Strains (b) Plasmids (c) Primers used for Strain Construction (d) Primers used for RT-qPCR (e) Primers used for ChIP (f) Primers used for Taq I-3C.
• MDAR checklist

### Data availability
All data generated or analysed during this study are included in the manuscript and supporting files.

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
