## [Editor Report · eLife assessment]

This is a **valuable** contribution to our understanding of how different cell stressors (ethanol or heat-shock) elicit unique responses at the genomic and topographical level under the regulation of yeast transcription factor Hsf1, providing **solid** evidence documenting the temporal coupling (or lack thereof) between Hsf1 aggregation and long-range communication among co-regulated heat-shock loci versus chromatin remodeling and transcriptional activation. A particular strength is the combination of genomic and imaging-based experimental approaches applied to genetically engineered in vivo systems.

---

## [Referee Report · Reviewer #2 (Public Review)]

Rubio et al. study the behavior of the transcription factor Hsf1 under ethanol stress, examining its distribution within the nucleus and the coalescence of heat shock response genes in budding yeast. In comparison to the heat shock response, the response to ethanol stress shows similar gene coalescence and Hsf1 binding. However, there is a notable delay in the transcriptional response to ethanol, and a disconnect between it and the appearance of irreversible Hsf1 condensates/puncta, highlighting important differences in how Hsf1 responds to these two related but distinct environmental stresses.

The authors have addressed the majority of my previous comments effectively. The Sis1 experiment provides a clear illustration of a distinctive response to ethanol and heat. This work offers a comprehensive perspective on Hsf1 in stress response from multiple angles.

---

## [Referee Report · Reviewer #3 (Public Review)]

This is an interesting manuscript that builds off of this group's previous work focused on the interface between Hsf1, heat shock protein (HSP) mRNA production, and 3D genome topology. Here the group subjects the yeast *Saccharomyces cerevisiae* to either heat stress (HS) or ethanol stress (ES) and examines Hsf1 and Pol II chromatin binding, Histone occupancy, Hsf1 condensates, HSP gene coalescence (by 3C and live cell imaging), and HSP mRNA expression (by RT-qPCR and live cell imaging). The manuscript is well written, and the experiments seem well done, and generally rigorous, with orthogonal approaches performed to support conclusions. The main findings are that both HS and ES result in Hsf1/Pol II-dependent intergenic interactions, along with formation of Hsf1 condensates. Yet, while HS results in rapid and strong induction of HSP gene expression and Hsf1 condensate resolution, ES result in slow and weak induction of HSP gene expression without Hsf1 condensate resolution. Thus, the conclusion is somewhat phenomenological - that the same transcription factor can drive distinct transcription, topologic, and phase-separation behavior in response to different types of stress.

---

## [Author Response]

The following is the authors’ response to the previous reviews.

**Reviewer #2 (Public Review):**
The authors have addressed the majority of my comments effectively. The new Sis1 experiment provides a clear illustration of a distinctive response to ethanol and heat. This work offers a comprehensive perspective on Hsf1 in stress response from multiple angles. I have two additional comments to improve the paper without re-review:(Original point #3) Could the authors clarify the differences between DPY1561 and the original strain used? There appears to be missing statistical analysis for Figure 1E at the bottom.

DPY1561 is a haploid version of the original heterozygous diploid strain (LRY033). We opted for this strain in the analysis depicted in Figures 1D and 1E since 100% of Hsp104 is BFP-tagged; thus, the signal above background is stronger and the scoring of Hsp104 foci cleaner. The statistical analysis (Mann Whitney test) for the lower graphs in Fig. 1E has been added. We thank the reviewer for pointing this out.

(Original point #4) In the new Figure 7F, '% transcription' and '% coalescence' are presented. My understanding is that Figures 7D and 7E aim to demonstrate the correlation between HSP104 transcription (a continuous variable) and HSP104-HSP12 coalescence (a binary variable) at the single-cell level. However, averaging the data across cells masks individual variations and potential anti-correlations. The authors could explore statistical methods that handle correlations between a continuous variable and a binary variable. Alternatively, consider converting 'HSP104 transcription' to a binary variable and then performing a chi-square test to assess the association.

We thank the reviewer for this suggestion. In response, we have made the following changes:

(1) Clarified that the data used in this analysis were derived from Fig. 7 – figure supplement 1 in which ‘HSP104 transcription’ was converted to a binary variable.

(2) Indicated that the theoretical ceiling for coalescence of these tagged alleles is 25% given their heterozygous state (Figure 7–figure supplement 1D legend). In the other 75% of cells scored, *HSP104-HSP12* coalescence might also be taking place but is not detectable using this strategy. Therefore, it is not possible to elucidate any anti-correlation between HSR transcription and HSR coalescence in this experiment.

In addition, we attempted to buttress the argument suggested by the Pearson correlation coefficient analysis (Fig. 7F) that a stronger association exists between transcription and gene coalescence in heat-shocked (HS) vs. ethanol stressed (ES) cells. To do so, we used the chi-square test as suggested by the reviewer. However, the results of this test were ambiguous, and we therefore did not include it in the manuscript.